# PAD: A Dataset and Benchmark for Pose-agnostic Anomaly Detection

**Qiang Zhou**[1*]   **Weize Li**[1*†]   **Lihan Jiang**[2†]   **Guoliang Wang**[1†]
**Guyue Zhou**[1]   **Shanghang Zhang**[3]   **Hao Zhao**[1]

[1]AIR, Tsinghua University   [2]Wuhan University   [3]National Key Laboratory for Multimedia
Information Processing, School of Computer Science, Peking University
{bamboosdu, liweize0224}@gmail.com   shanghang@pku.edu.cn
zhaohao@air.tsinghua.edu.cn

## Abstract

Object anomaly detection is an important problem in the field of machine vision
and has seen remarkable progress recently. However, two significant challenges
hinder its research and application. First, existing datasets lack comprehensive
visual information from various pose angles. They usually have an unrealistic
assumption that the anomaly-free training dataset is pose-aligned, and the testing
samples have the same pose as the training data. However, in practice, anomaly
may exist in any regions on a object, the training and query samples may have
different poses, calling for the study on pose-agnostic anomaly detection. Second,
the absence of a consensus on experimental protocols for pose-agnostic anomaly de-
tection leads to unfair comparisons of different methods, hindering the research on
pose-agnostic anomaly detection. To address these issues, we develop Multi-pose
Anomaly Detection (MAD) dataset and Pose-agnostic Anomaly Detection (PAD)
benchmark, which takes the first step to address the pose-agnostic anomaly detec-
tion problem. Specifically, we build MAD using 20 complex-shaped LEGO toys
including 4K views with various poses, and high-quality and diverse 3D anomalies
in both simulated and real environments. Additionally, we propose a novel method
OmniposeAD, trained using MAD, specifically designed for pose-agnostic anomaly
detection. Through comprehensive evaluations, we demonstrate the relevance of
our dataset and method. Furthermore, we provide an open-source benchmark
library, including dataset and baseline methods that cover 8 anomaly detection
paradigms, to facilitate future research and application in this domain. Code, data,
and models are publicly available at https://github.com/EricLee0224/PAD.

## 1   Introduction

Unsupervised visual anomaly detection attempts to detect shape or texture anomalies using normal
samples and representations from pre-trained model. Due to the increasing demand for quality
control of manufactured products and the complicated shape of object and its texture information,
object-centric anomaly detection has attracted growing interest. However, existing object anomaly
detection datasets and methods are designed under the pose-aligned assumption, which **limits the
ability to detect potentially occluded anomalies from arbitrary pose views.** By overcoming
the inherent constraints of pose-aligned anomaly detection approaches, the pose-agnostic anomaly
detection setting offers enhanced flexibility and applicability in real-world scenarios where a object
may exhibit anomalies from diverse perspectives.

---

[*]Equal contribution.
[†]Work done during internship at AIR.

37th Conference on Neural Information Processing Systems (NeurIPS 2023) Track on Datasets and Benchmarks.

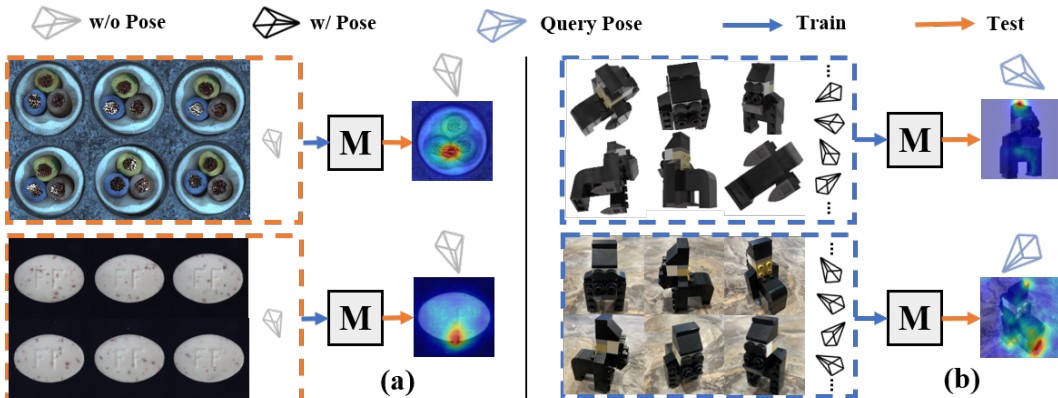

Figure 1: (a) Existing assumption for Pose-aligned Anomaly Detection; (b) Our introduced Pose-agnostic Anomaly Detection setting. **M** denotes pre-trained AD model.

To address the pose-agnostic anomaly detection problem, two key challenges need to be overcome. First, we currently lack datasets that simultaneously provide RGB images of object's multiple viewpoints and its pose labels. This limitation hinders the model's ability to learn the decision boundaries in high-level feature space of anomaly-free object in terms of both shape and texture, which is critical for performing accurate pose-agnostic anomaly detection. Second, in addition to above mentioned dataset being needed, how to measure and compare the performance of pose-agnostic anomaly detection models remains an issue. Existing benchmarks focus on pose-aligned anomaly detection, making it difficult to establish fair criterion for evaluating the effectiveness of pose-agnostic anomaly detection methods.

To this end, we developed the multi-pose anomaly detection (MAD) dataset that contributes to a comprehensive evaluation of pose-agnostic anomaly detection. Our dataset consists of 20 diverse LEGO toys, capturing more than 11,000 images from different viewpoints covering a wide range of poses. Furthermore, the dataset includes three types of anomalies that was carefully selected to reflect both the simulated and real-world environments. To explore our MAD dataset, we make in-depth analyses from two perspectives (shape and texture) with interesting observations. We conducted the analysis of the shape and texture complexity for different categories of LEGO toys in the dataset to gain insight into the performance differences of the baseline methods. By investigating the correlation between model performance and attribute complexity, we aim to assess whether the baseline methods are effective in capturing the full range of visual information from normal objects. Alongside the dataset, we propose OmniposeAD, which introduces neural radiance field into the anomaly detection paradigm for the first time. It overcomes the difficulty of learning the decision boundaries of a normal object from both shape and texture, has the ability to detect anomalies in multiple views of an object, and is even able to detect anomalies in poses that have never been seen in the training set. The main contributions are summarized as follows:

- We introduced **P**ose-agnostic **A**nomaly **D**etection (PAD), a challenging setting for object anomaly detection from diverse viewpoints/poses, breaking free from the constraints of stereotypical pose alignment, and taking a step forward to pratical anomaly detection and localization tasks.

- We developed the **M**ulti-pose **A**nomaly **D**etection (MAD) dataset, the first attempt at evaluating the pose-agnostic anomaly detection. Our dataset comprises MAD-Sim and MAD-Real, containing 11,000+ high-resolution RGB images of multi-pose views from 20 diverse shapes and colors of LEGO toys, offering pixel-precise ground truth for 3 types of anomalies.

- We conducted a comprehensive benchmark of the PAD setting, utilizing the MAD dataset. Across 8 distinct paradigms, we meticulously selected 10 state-of-the-art methods for evaluation. In a pioneering effort, we delving into the previously unexplored realm of correlation between object attributes and anomaly detection performance.

- We proposed ***OmniposeAD***, utilizing NeRF to encode anomaly-free objects from diverse viewpoints/poses and comparing reconstructed normal reference with query image for pose-agnostic anomaly detection and localization. It outperforms previous methods quantitatively and qualitatively on the MAD benchmark, which indicates the promises of PAD setting.

Table 1: Comparison of MAD and existing object anomaly detection datasets. Abbreviated letters refer to the details[1].

| Datasets | Year. | Type. | Represent. | Sample statistics | | | Object Attributes | | |
|---|---|---|---|---|---|---|---|---|---|
| | | | | #Cls. | #Normal | #Abnormal | Color | Structure | Pose |
| GDXray[25] | 2015 | Real | RGB-D | 1 | 0 | 19,407 | Grayscale | C | ✗ |
| *MVTec:* | | | | | | | | | |
| *AD[5]* | 2019 | Real | RGB | 15 | 4,096 | 1,258 | Diverse | S | ✗ |
| *3D AD[7]* | 2021 | Real | RGB/PC | 15 | 2,904 | 948 | Diverse | S | ✗ |
| *LOCO-AD[4]* | 2022 | Real | RGB | 5 | 2,347 | 993 | Diverse | C | ✗ |
| MPDD[20] | 2021 | Real | RGB | 6 | 1,064 | 282 | Diverse | C | ✗ |
| Eyecandies[8] | 2022 | Syn. | RGB/D/N | 10 | 13,250 | 2,250 | Diverse | S | ✗ |
| VisA[65] | 2022 | Real | RGB | 12 | 9,621 | 1,200 | Diverse | S&C | ✗ |
| **MAD (Ours)** | 2023 | Sim+Real | RGB | 20 | 5,231 | 4,902 | Diverse | S&C | ✓ |

## 2 Related Work

**Object Anomaly Detection Datasets.** The progress of object anomaly detection in industrial vision is significantly impeded by the scarcity of datasets containing high-quality annotated anomaly samples and comprehensive view information about normal objects. MVTec has developed a series of widely-used photo-realistic industrial anomaly detection datasets: The objects provided by the AD[5] dataset are simple, as discerning anomalies can be achieved solely from a single view. Although the 3D-AD[7] dataset offers more complex objects, it lacks RGB information from a full range of views, requiring the supplementation of hard-to-capture point cloud data to detect subtle structural anomalies. The LOCO AD[4] dataset provides rich global structural and logical information but is not suitable for fine-grained anomaly detection on individual objects. GDXray[25] provides grayscale maps obtained through X-ray scans for visual discrimination of structural defects but lacks normal samples and color/texture information. The MPDD[20] dataset offers multi-angle information about the objects but is limited in size and lacks standardized backgrounds in the photos. Recently, Eyecandies[8] has introduced a substantial collection of synthetic candy views captured under various lighting conditions and provides multi sensory inputs. However, there remains a significant gap between synthesized data and the real or simulated data domain. VisA[65] contains objects with complex structures, multiple instances at different locations in single view, different objects across 3 domains, and multiple anomaly classes, thus is closer to the real world than MvTec-AD[5]. To address these issues and enable exploration of the pose-agnostic AD problem, we propose Multi-pose Anomaly Detection (MAD) dataset. We present comprehensive comparison between MAD and other representative object anomaly detection datasets in Table.1.

**Unsupervised Anomaly Detection for Visual Inspection.** Existing methods can be categorized into feature embedding-based and reconstruction-based strategies, according to a recent survey[24]. Feature embedding-based methods such as Teacher-Student Architecture[6, 51, 15, 10, 35], One-Class Classification[17, 56, 62, 31, 22, 54], Distribution-map[36, 42, 33, 34, 18, 59, 44], and Memory Bank[32, 13, 21, 2, 50] aim to learn low-dimensional representations of input data to capture underlying patterns and anomalies. Reconstruction-based methods exploiting Autoencoder (AE)[3, 53, 61, 60, 14, 46], Generative Adversarial Networks (GANs)[38, 37, 52, 39, 23], and Transformer[27, 58, 29, 57] model aim to learn a representation that can effectively reconstruct normal samples. Deviations in reconstruction quality indicate the presence of anomalies. BTF[19] reveals that using only angularly limited RGB images to determine whether an object is defective or not can easily overlook some obscured structural anomalies. Therefore it is necessary to develop methods for 3D anomaly detection using 3D information, such as point clouds, depth, etc. AST[35] employs RGB image with depth information to enhance anomaly detection performance. M3DM[47] and CPMF[11] encourage the fusion of features from different modalities in RGB and point cloud information, and combine the local geometric information of the 3D modal with the global semantic information of the pseudo 2D modal.

**Neural Radiance Field.** Neural Radiation Field (NeRF)[26] implements an implicit representation of the scene, receiving hundreds of sampling points along each camera ray and outputting the predicted color and density. Because NeRF can handle complex illumination and occlusion situations, we can use it to capture the color and shape properties of objects in real scenes, which means that NeRF has

---

[1]Syn., PC, D, N, C and S denotes Synthesis, Point Cloud, Depth, Normal Map, Complex, and Simple, respectively.

the potential to be applied to anomaly detection tasks. It has also been used for numerous applications, such as large-scale urban reconstruction[48, 49, 40], human face or body reconstruction[55, 12], and robotic applications such as pose estimation and SLAM[41, 64, 63]. Since NeRF allows for the continuous modeling of 3D scenes and the re-rendering of high-quality photos, it can create reference images from any angle for anomaly detection in this work.

## 3 MAD: Multi-pose Anomaly Detection Dataset

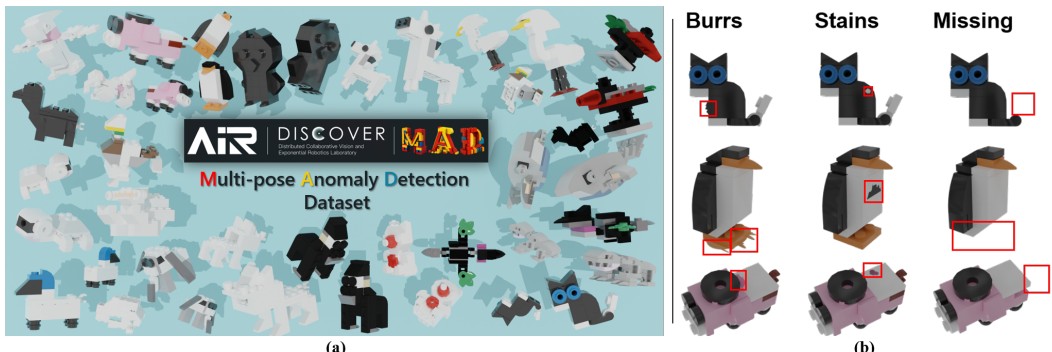

Figure 2: (a) Our LEGO family in MAD dataset; (b) Examples of three defect types.

### 3.1 Data Selection

Aiming to support pose-agnostic anomaly detection, our dataset selects 20 LEGO animals toys with diverse shape complexity and color contrasts as target objects shown in Figure.2(a). These selected toys encompass a wide range of shapes and sizes, ranging from *cat* and *pig* to larger animals like *unicorn* and *elephant*, along with other interesting animal forms. Each animal toy is meticulously designed and constructed to ensure a variety of shapes and complexities. In selecting these objects, we also paid attention to color contrasts. Bright and vibrant colors are used in the construction of each animal toy, making them more visually striking and easily recognizable in the images. The choice of color contrasts aims to assist algorithms in accurately detecting and localizing anomalies in different environments and backgrounds.

### 3.2 Data Processing and Annotation

We used Blender in combination with Ldrew (LEGO parts library) to build 20 different types of 3D LEGO animal models and generate anomalous such as burrs, stains and missing parts shown as Figure.2(b). To achieve a full pose angle rendering of the normal samples, we established the center of the animal as the origin and positioned the rendering camera around the sphere. By maintaining a fixed radius, we generated multiple cameras simultaneously, each positioned at a different theta/phi coordinate in the spherical coordinate system. In addition, we created ten cameras equally spaced along the Z-axis, with the camera orientation set to face the animal. In order to edit the real anomalies on the simulation model, we hired professional LEGO players to manually modify the 20 types of LEGO animal models according to the types of anomalies encountered using PhotoShop. For the hand-edited anomalies, synchronized ground truth labels were produced by a computer vision researcher. Notably, only a subset of images showed the anomaly portion, which required a manual selection process to determine the appropriate data for anomaly detection. Ultimately, approximately150-300 anomaly images were generated for each animal. To generate training data with more normal instances, we used the same approach. For the real dataset, we collect parts from the actual production line, including abnormal parts characterized by real burrs and stains. In addition, we manually generated the missing-piece anomaly by removing LEGO parts. Given the rarity of anomalies in the manufacturing process of LEGO toys and the difficulty in obtaining these parts, we deliberately reduced the sample size of the real-world version.

### 3.3 Data Statistics and Split

We present the statistical information of the dataset in Table.1. During the data spliting process, we followed design strategy of existing anomaly detection datasets. MAD-Sim is a dataset created programmatically using Blender, allowing for controlled sample quantities. We divided it into training and test sets with a sample ratio of approximately 2:1 to explore anomaly detection algorithm performance on multi-view objects. For MAD-Real, the data was captured in real-world scenes using a camera. Anomalies are infrequent in actual production, and we aimed to reflect this reality. MAD-Real has approximately one-fifth the total sample count of MAD-Sim and follows a similar division into training and test sets, maintaining a sample ratio of about 4:1.

### 3.4 Object Attributes Quantification

Anomaly detection algorithms are evolving rapidly, but its often difficult to intuitively determine which one is the best from performance metrics when trying to deploy an algorithm to a real scenario. To reduce testing costs, we explore a novel aspect in anomaly detection tasks, which is the relationship between object attributes and anomaly detection performance. When we are given an object to be tested, we expect to determine which algorithm is likely to be the best, simply by referring to the correlation between quantitative metrics of the object's attributes, i.e., object shape complexity and color contrast, and the performance of different algorithms. The quantitative details are shown in Fig.3.

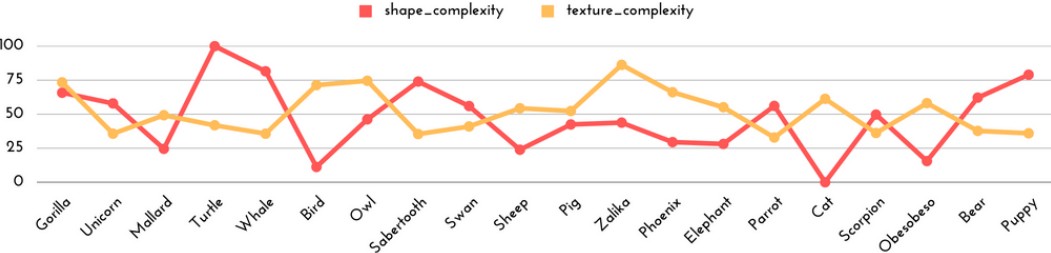

Figure 3: Object attributes quantification of 20 LEGO toys.

## 4 Pose-agnostic Anomaly Detection Framework

### 4.1 Problem Definition

Our Pose-agnostic Anomaly Detection (PAD) setting introduced for object anomaly detection and localization tasks is shown as Figure.1 and it can be formally stated as follow: Given a training set $\mathcal{T} = \{t_i\}_{i=1}^{N}$, in which $\{t_1, t_2, \cdots, t_N\}$ are the anomaly-free samples from object's multi pose view and each sample $t$ consists of a RGB image $I_{\mathrm{rgb}}$ w/ pose information $\theta_{\mathrm{pose}}$. In addition, $\mathcal{T}$ belongs to certain object $o_j$, $o_j \in \mathcal{O}$, where $\mathcal{O}$ denotes the set of all objects categories. During testing, given a query (normal or abnormal) image $\mathcal{Q}$ from object $o_j$ w/o pose information $\theta_{\mathrm{pose}}$, the pre-trained AD model $M$ should discriminate whether or not the query image $\mathcal{Q}$ is anomalous and localize the pixel-wise anomaly region if the anomaly is detected.

### 4.2 OmniposeAD

The OmniposeAD illustrated in Figure.4, consists of anomaly-free neural radiance field, coarse-to-fine pose estimation module, and anomaly detection and localization module. The input of OmniposeAD is query image $\mathcal{Q}$ w/o pose. Initially, $\mathcal{Q}$ undergoes the coarse-to-fine pose estimation module to obtain the accurate camera view pose $\hat{\theta}_{\mathrm{pose}}$. Subsequently, $\hat{\theta}_{\mathrm{pose}}$ is utilized in the pre-trained neural radiance field for rendering the normal reference. Finally, the reference is compared to the $\mathcal{Q}$ to extract the anomaly information.

#### 4.2.1 Anomaly-free reference view synthesis

To reconstruct anomaly-free references from various viewpoints, we employ unsupervised algorithm capable of synthesizing novel views. With an anomaly-free training set $\mathcal{T}$, we represent anomaly-free

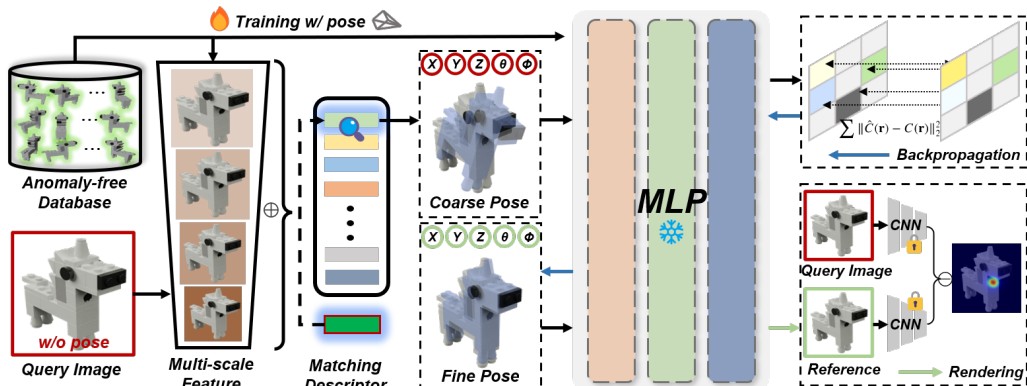

Figure 4: OmniposeAD consists of three componets. 1) The Neural Radiance Field is trained using multi-pose views of anomaly-free objects to capture normal information. 2) The query image proceeds through two-stage pose estimation process, from coarse to fine: a matching descriptor is generated to match with the anomaly-free database samples to obtain coarse pose, then the pose refined using the iNeRF[55] to obtain the reference pose. 3) The pre-trained Neural Radiance Field renders the normal reference, which is compared with the query image for anomaly detection and localization.

objects implicitly using vanilla neural radiance field, following [26] which employs a multi layer perceptron (MLP) network and minimizes the photometric loss.

### 4.2.2 Coarse-to-fine pose estimation

To generate normal reference using the pre-trained neural radiance field, the fine camera pose $\hat{\theta}_{\text{pose}}$ of the query image needs to be determined. For this purpose, we employ the coarse-to-fine pose estimation module that combines image retrieval techniques and iNeRF[55].

**On a coarse scale** Our objective is to roughly align the query image $\mathcal{Q}$ with an anomaly-free image with camera pose in the training database i.e., find a function $\phi$ that satisfies the following criteria:

$$\phi : \mathbb{R}^{H \times W \times 3} \mapsto \mathbb{R}^{64} \quad \phi(\mathcal{Q}) = \mathbf{q} \in \mathbb{R}^{64} \tag{1}$$

First, we utilize the pre-trained EfficientNet-B4 model[43] as the backbone for feature extraction. Following the standard procedure[58], we input the multi-scale feature map into the network to obtain the descriptor $\mathbf{q}$. Given an 3-channel RGB query image $I_{\text{rgb}}$ of dimension $H \times W$, our model $\phi$ extract a compact 64-dimensional (64$\mathcal{D}$) descriptor or embedding $\phi(x)$. Then the image retrieval task[9, 1, 30] considers a database of images $I_{\text{rgb}}$ for each sample $t_i$ in the training set $\mathcal{T}$, and given query image $\mathcal{Q}$, we calculate:

$$\underset{\mathcal{T}}{\text{argmin}} \|\phi(\mathcal{Q}) - \phi(I)\|_2^2 \tag{2}$$

Finally, the top-1 most similar image $I_{\text{rgb}}$ is retrieved. We utilize the associated pose $\theta_{\text{pose}}$ of the image $I_{\text{rgb}}$ as the initial coarse pose $\hat{\theta}_{\text{pose}}$ for the next stage.

**On a fine scale** We refine the estimated pose using iNeRF[55]. This method enables us to fully utilize the pre-trained NeRF model and estimate camera poses. iNeRF uses the ability from NeRF to take some estimated camera pose $\theta \in \text{SE}(3)$ in the coordinate frame of the NeRF model and render a corresponding image observation, and then use the same photometric loss function as NeRF but updates the fine pose $\hat{\theta}_{\text{pose}}$ instead of the weights of the MLP network.

### 4.2.3 Anomaly detection and localization

When the normal reference is rendered using $\hat{\theta}_{\text{pose}}$ and query image $\mathcal{Q}$ are available, we use the frozen pre-trained CNN backbone for feature extraction. The feature from layer1 to layer5 are resized

to the same size as the query image. Then the multi-scale feature and the query image are concatenate together to form a feature map, $q \in R^{C \times H \times W}$ with the corresponding input pixel. The query image corresponds to $f_q \in R^{C \times H \times W}$, while the rendered reference corresponds to $\hat{f} \in R^{C \times H \times W}$. We first define the feature difference map, $d(i, u)$, as shown in Eqn. 3

$$\boldsymbol{d}(i, u) = \boldsymbol{f_q}(i, u) - \hat{\boldsymbol{f}}(i, u), \tag{3}$$

where $i$ denotes the channel index and, $u$ is the spatial position index (height together with width for simplicity).

$$\boldsymbol{s}(u) = ||\boldsymbol{d}(:, u)||_2. \tag{4}$$

**Anomaly detection** aims to determine if an image contains anomalous regions. We intuitively use the maximum value of the averagely pooled $\boldsymbol{s}(u)$ as the anomaly score of the whole image.

**Anomaly localization** aims to localize anomalous regions, producing an anomaly score map, $\boldsymbol{s}(u)$ in Eqn. 4 , which assigns an anomaly score for each pixel, $u$. $\boldsymbol{s}(u)$ is calculated as the $L2$ norm of the feature difference vector, $\boldsymbol{d}(:, u)$ in Eqn. 3.

Table 2: Quantitative results of pixel/image-level anomaly detection performance on MAD.

| Category | Feature Embedding-based | | | | | | Reconstruction-based | | | | Ours |
|---|---|---|---|---|---|---|---|---|---|---|---|
| | Patchcore[32] | STFPM[45] | Fastflow[59] | CFlow[18] | CFA[21] | Cutpaste[22] | DRAEM[16] | FAVAE[51] | OCRGAN[28] | UniAD[57] | |
| Gorilla | 88.4/66.8 | 93.8/65.3 | 91.4/51.1 | 94.7/69.2 | 91.4/41.8 | 36.1/- | 77.7/58.9 | 92.1/46.8 | 94.2/- | 93.4/56.6 | **99.5/93.6** |
| Unicorn | 58.9/92.4 | 89.3/79.6 | 77.9/45.0 | 89.9/82.3 | 85.2/85.6 | 69.6/- | 26.0/70.4 | 88.0/68.3 | 86.7/- | 86.8/73.0 | **98.2/94.0** |
| Mallard | 66.1/59.3 | 86.0/42.2 | 85.0/72.1 | 87.3/74.9 | 83.7/36.6 | 40.9/- | 47.8/34.5 | 85.3/33.6 | 88.9/- | 85.4/70.0 | **97.4/84.7** |
| Turtle | 77.5/87.0 | 91.0/64.4 | 83.9/67.7 | 90.2/51.0 | 88.7/58.3 | 77.2/- | 45.3/18.4 | 89.9/82.8 | 76.7/- | 88.9/50.2 | **99.1/95.6** |
| Whale | 60.9/**86.0** | 88.6/64.1 | 86.5/53.2 | 89.2/57.0 | 87.9/77.7 | 66.8/- | 55.9/65.8 | 90.1/62.5 | 89.4/- | 90.7/75.5 | **98.3**/82.5 |
| Bird | 88.6/82.9 | 90.6/52.4 | 90.4/76.5 | 91.8/75.6 | 92.2/78.4 | 71.7/- | 60.3/69.1 | 91.6/73.3 | **99.1**/- | 91.1/74.7 | 95.7/**92.4** |
| Owl | 86.3/72.9 | 91.8/72.7 | 90.7/58.2 | 94.6/76.5 | 93.9/74.0 | 51.9/- | 78.9/67.2 | 96.7/62.5 | 90.1/- | 92.8/65.3 | **99.4/88.2** |
| Sabertooth | 69.4/76.6 | 89.3/56.0 | 88.7/70.5 | 93.3/71.3 | 88.0/64.2 | 71.2/- | 26.2/68.6 | 94.5/82.4 | 91.7/- | 90.3/61.2 | **98.5/95.7** |
| Swan | 73.5/75.2 | 90.8/53.6 | 89.5/63.9 | 93.1/67.4 | 95.0/66.7 | 57.2/- | 75.9/59.7 | 87.4/50.6 | 72.2/- | 90.6/57.5 | **98.8/86.5** |
| Sheep | 79.9/89.4 | 93.2/56.5 | 91.0/71.4 | 94.3/80.9 | 94.1/86.5 | 67.2/- | 70.5/59.5 | 94.3/74.9 | **98.9**/- | 92.9/70.4 | 97.7/**90.1** |
| Pig | 83.5/85.7 | 94.2/50.6 | 93.6/59.6 | 97.1/72.1 | 95.6/66.7 | 52.3/- | 65.6/64.4 | 92.2/52.5 | 93.6/- | 94.8/54.6 | **97.7/88.3** |
| Zalika | 64.9/68.2 | 86.2/53.7 | 84.6/54.9 | 89.4/66.9 | 87.7/52.1 | 43.5/- | 66.6/51.7 | 86.4/34.6 | 94.4/- | 86.7/50.5 | **99.1/88.2** |
| Phoenix | 62.4/71.4 | 86.1/56.7 | 85.7/53.4 | 87.3/64.4 | 87.0/65.9 | 53.1/- | 38.7/53.1 | 92.4/65.2 | 86.8/- | 84.7/55.4 | **99.4/82.3** |
| Elephant | 56.2/78.6 | 76.8/61.7 | 76.8/61.6 | 72.4/70.1 | 77.8/71.7 | 56.9/- | 55.9/62.5 | 72.0/49.1 | 91.7/- | 70.7/59.3 | **99.0/92.5** |
| Parrot | 70.7/78.0 | 84.0/61.1 | 84.0/53.4 | 86.8/67.9 | 83.7/69.8 | 55.4/- | 34.4/62.3 | 87.7/46.1 | 66.5/- | 85.6/53.4 | **99.5/97.0** |
| Cat | 85.6/78.7 | 93.7/52.2 | 93.7/51.3 | 94.7/65.8 | 95.0/68.2 | 58.3/- | 79.4/61.3 | 94.0/53.2 | 91.3/- | 93.8/53.1 | **97.7/84.9** |
| Scorpion | 79.9/82.1 | 90.7/68.9 | 74.3/51.9 | 91.9/79.5 | 92.2/91.4 | 71.2/- | 79.7/83.7 | 88.4/66.9 | **97.6**/- | 92.2/69.5 | 95.9/**91.5** |
| Obesobeso | 91.9/89.5 | 94.2/60.8 | 92.9/67.6 | 95.8/80.0 | 96.2/80.6 | 73.3/- | 89.2/73.9 | 92.7/58.2 | **98.5**/- | 93.6/67.7 | 98.0/**97.1** |
| Bear | 79.5/84.2 | 90.6/60.7 | 82.8/72.9 | 92.2/81.4 | 90.7/78.7 | 68.8/- | 39.2/76.1 | 90.1/52.8 | 83.1/- | 90.9/65.1 | **99.3/98.8** |
| Puppy | 73.3/65.6 | 84.9/56.7 | 80.3/59.5 | 89.6/71.4 | 82.3/53.7 | 43.2/- | 45.8/57.4 | 85.6/43.5 | 78.9/- | 87.1/55.6 | **98.8/93.5** |
| Mean | 74.7/78.5 | 89.3/59.5 | 86.1/60.8 | 90.8/71.3 | 89.8/68.2 | 59.3/- | 58.0/60.9 | 89.4/58.0 | 88.5/- | 89.1/62.2 | **97.8/90.9** |

## 5 Experiments and Analysis

Our experiments aim to broadly evaluate the performance of anomaly detection algorithms in a pose-agnostic setting, where different types of anomalies may appear in different object poses. We selected representative state-of-the-art methods from different paradigms for fair preliminary benchmarking using the MAD dataset we developed with the aim of exploring the applicability of existing methods in the PAD setting. In addition, we evaluate the performance of our proposed OmniposeAD, normal reference reconstruction results, and the effect of training poses/viewpoints richness on performance.

**Benchmark Methods Selection.** The selection criteria for benchmark methods include representativeness, superior performance, and availability of source code. To comprehensively investigate the performance of anomaly detection algorithms in the pose-agnostic anomaly detection setting, we selected representative methods from 8 paradigms, the taxonomy can be found in the supplementary. For the Feature Embedding-based strategy, we selected Patchcore[32], STFPM[45], Fastflow[59], CFlow[18], CFA[21], and Cutpaste[22]. For the Reconstruction-based strategy, we selected DRAEM[16], FAVAE[51], OCR-GAN[28], and UniAD[57]. It is important to note that 3D anomaly detection methods utilizing multimodal information were not considered in order to ensure a fair comparison, as the MAD dataset only provides RGB information.

**Implementation Details.** For the implementation details, we trained the OmniposeAD using the MAD dataset, employing the same hyperparameters for both synthetic and real scenes. During the anomaly-free novel view synthesis phase, the process took approximately 5 hours with a batch size of 4096. The image resolution was set to 400x400 for the simulated dataset and 504x378 for the

real dataset to ensure efficient processing. Additionally, the coarse-to-fine pose estimation process involved a total of 300 optimization steps. We initialized the learning rate to 0.01 and decayed it to 5.8e-05 at epoch 50 to facilitate a smooth optimization process, utilizing a batch size of 3072. The entire framework required approximately 10-15 hours to run on a single NVIDIA Tesla A100.

**Evaluation Metric.** Following previous work, we specifically choose the Area Under the Receiver Operating Caracteristic Curve (AUROC) as the primary metric for evaluating the performance of anomaly segmentation at the pixel-level and anomaly classification at the image-level. While there exist various evaluation metric for these tasks, AUROC stands out as the most widely used and suitable metric for conducting comprehensive benchmarking. The AUROC score can be calculated as follows:

$$\text{AUROC} = \int (\text{R}_{\text{TP}}) d\text{R}_{\text{FP}} \tag{5}$$

Here, $\text{R}_{\text{TP}}$ and $\text{R}_{\text{FP}}$ represent the pixel/image-level true positive rate and false positive rate, respectively.

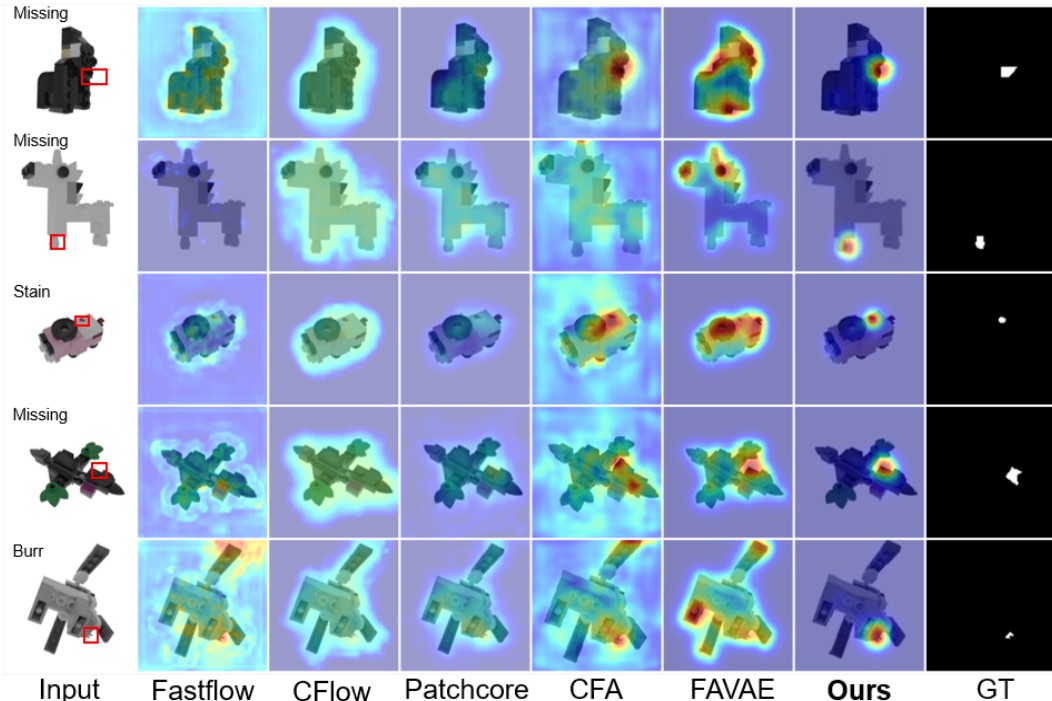

Figure 5: Qualitative results visualization of anomaly localization performance on MAD-Sim.

## 5.1 Pose-agnostic Anomaly Detection Benchmark on MAD Dataset

Our experiments provide a fair comparison between the state-of-the-art methods and OmniposeAD in the PAD setting, including pixel and image-level AUROC performance. Table 2 and Figure 5 clearly demonstrate that OmniposeAD achieves the state-of-the-art performance on MAD dataset for pose-agnostic anomaly detection evaluation. Specifically, OmniposeAD outperforms Feature Embedding-based CFlow[18] and Reconstruction-based UniAD[57] by a large margin, +7.0/+19.6 and +8.7/+28.7, respectively. Its noteworthy that additional data (i.e., using a pre-trained model on ImageNet during the training phase) always brings advantage. This consistent superiority can be attributed to ImageNet's extensive repository of object representations spanning diverse poses. In a direct comparison with methods not utilizing additional training data, our approach showcases a remarkable enhancement, as evidenced by a substantial increment of +29.2/+8.3. Both the Feature Embedding and Reconstruction-based methods, namely DRAEM[60] and Cutpaste[22], employ the generation of pseudo-anomalies as a data augmentation technique. However, these approaches

exhibit a notable decline in performance within the PAD setting. Objects may present different representations in different poses, leading to complex patterns of visual anomalies. It is difficult to effectively characterize the anomalies of objects under different viewpoints through simple data enhancement methods.

Our benchmarks demonstrate the considerable potential of both feature embedding-based and reconstruction-based approaches in the PAD setting. In particular, for the pixel-level anomaly localization task, the performance of STFPM[45](89.3), CFlow[18](90.8), CFA[21](89.8), FAVAE[51](89.4), and UniAD[57](89.1) shows a slightly fluctuating but largely respectable performance. However, with the exception of our proposed OmniposeAD, the performance of benchmark methods in image-level anomaly detection tasks is not satisfactory. It is worth noting that UniAD[57], although capable of learning multi-category anomaly boundaries using a single model, still struggles to achieve satisfactory image-level anomaly detection performance for multi-pose anomalies in the PAD setting. To address this gap, algorithms that are adept at capturing the normal attributes of multi-pose objects need to be explored to achieve reasonable image-level anomaly detection performance within the PAD setting.

## 5.2 Correlation of performance with object attributes

The results are shown in Figure.6, where the performance of most methods is positively correlated with color contrast and negatively correlated with structure complexity, which is consistent with our intuition. Notably, Cutpaste[22], a representative approach that generates anomalies and reconstructs them through a self-supervised task, stands out as being sensitive to color contrast but surprisingly tolerant towards shape complexity. Furthermore, we demonstrate the robustness of our proposed OmniposeAD to changes in object attributes.

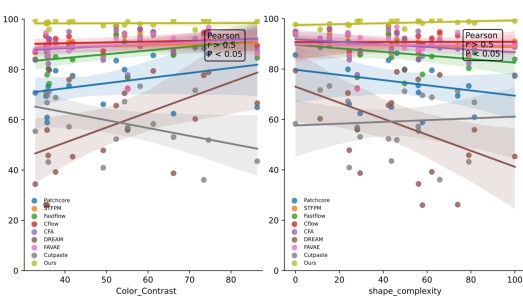

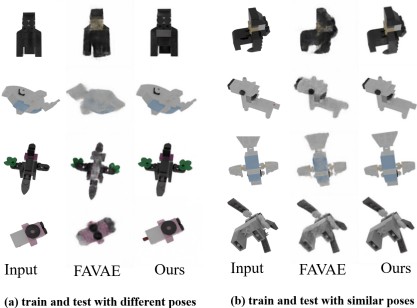

Figure 6: Correlation between object attributes and anomaly detection performance.

Figure 7: Visualization of reference reconstruction between OmniposeAD and FAVAE.

## 5.3 Validation of reference reconstruction on *OmniposeAD*

Both the reconstruction-based method and our proposed OmniposeAD essentially reconstruct the normal reference. In the quantitative results of the anomaly localization task, the representative reconstruction-based method FAVAE achieved 89.4, and we further visualized the reconstructed references to explore the gap. Figure.7 shows that FAVAE does not reconstruct references for unseen poses well (the quantitative results do not align with visual facts), hinting that the existing evaluation metrics are potentially unsatisfactory in the PAD setting. Our further discussion is provided in Sec.4 of the Supplementary.

## 5.4 Effect of Dense-to-Sparse training data on *OmniposeAD* performance

In this section, we utilize the MAD-Real dataset to evaluate the impact of the transition from dense to sparse viewpoint training data on the performance of the OmniposeAD method, reflecting the lack of available viewpoints for real-world applications. We control the sample size of the training set for the OmniposeAD model to be 100, 70, and 50, respectively. As shown in Table 3: Despite the sparse training data leads to a slight decrease in performance, the less pronounced drop in pixel-

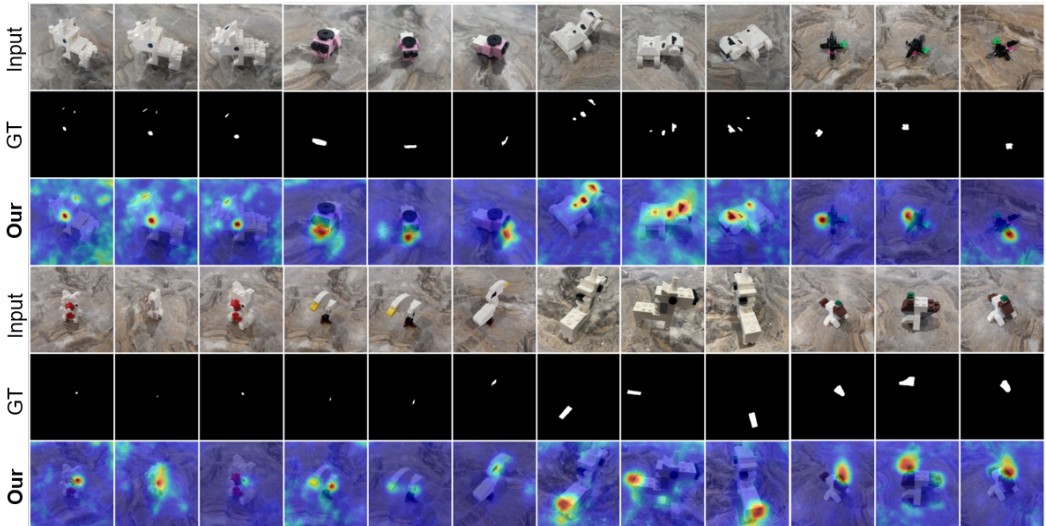

Figure 8: Qualitative results visualization of OmniposeAD anomaly localization performance from multi-view on MAD-Real.

level performance suggests that OmniposeAD excels in capturing local consistency within objects. The more significant decrease at the image-level indicates that sparse data impacts OmniposeAD's ability to learn the holistic normal features of object from global perspective. We further showed the multi-view anomaly localization performance of OmniposeAD on the MAD-Real dataset in Figure.8.

Table 3: Dense-view to sparse-view object anomaly detection results on MAD-Real.

| Sample Size | 100 | | 70 | | 50 | | Sample Size | 100 | | 70 | | 50 | |
| Object Class | Image | Pixel | Image | Pixel | Image | Pixel | Object Class | Image | Pixel | Image | Pixel | Image | Pixel |
| --- | --- | --- | --- | --- | --- | --- | --- | --- | --- | --- | --- | --- | --- |
| Gorilla | **93.6** | **99.5** | 85.4 | 97.5 | 91.0 | 97.4 | Pig | **75.7** | **96.4** | 73.9 | 95.7 | 72.8 | 95.4 |
| Unicorn | 86.2 | **96.5** | 84.4 | 96.1 | 80.2 | 94.9 | Zalika | 88.2 | **99.1** | 78.8 | 96.8 | 72.8 | 96.0 |
| Mallard | 87.3 | **96.3** | 80.3 | 94.4 | 74.6 | 95.1 | Phoenix | 86.0 | **99.4** | 84.6 | 99.2 | 82.6 | 98.8 |
| Turtle | **99.1** | 95.4 | 95.1 | 92.3 | 83.4 | 90.4 | Elephant | **91.0** | **98.1** | 82.5 | 95.9 | 85.1 | 96.5 |
| Whale | 82.1 | **98.5** | 82.5 | 97.0 | 76.0 | 93.1 | Parrot | 91.5 | **99.1** | 85.5 | 98.2 | 87.9 | 96.9 |
| Bird | 92.0 | 95.1 | 91.6 | 91.7 | 89.1 | 93.4 | Cat | 78.9 | **97.6** | 73.5 | 97.6 | 73.7 | 97.1 |
| Owl | 89.1 | **99.3** | 78.8 | 98.1 | 89.1 | 98.8 | Scorpion | 87.4 | **94.2** | 77.0 | 91.8 | 74.3 | 91.5 |
| Sabertooth | **93.8** | 97.8 | 84.9 | 95.1 | 81.5 | 94.4 | Obesobeso | 88.5 | **96.1** | 86.4 | 97.9 | 81.8 | 97.8 |
| Swan | 80.1 | **98.0** | 77.0 | 97.3 | 71.3 | 96.7 | Bear | 97.8 | **99.3** | 92.0 | 98.0 | 87.8 | 97.9 |
| Sheep | **84.3** | **97.6** | 83.2 | 97.6 | 82.3 | 92.8 | Puppy | **93.1** | **98.5** | 86.5 | 95.8 | 80.3 | 95.3 |

## 6   Discussion

**Conclusion.** In this work, we introduce pose-agnostic anomaly detection setting, and develop the MAD dataset, which is the first exploration to evaluate pose-agnostic anomaly detection algorithms. In addition, we propose OmniposeAD to solve the problem of unsupervised anomaly detection in different poses of an object, which avoids missing potential occluded anomaly regions. **Limitation.** For the MAD dataset, which includes only standard rigid objects and no naturally varying objects, the complexity and diversity of industrial products made it difficult to collect and quantify a complete sample of diverse features. For OmniposeAD, it is difficult to generalize to novel classes to obtain appreciable performance; for all modules, we chose the vanilla method in pursuit of robustness, which has the potential for faster training times, less training data, and more realistic high-frequency details.

## 7   Acknowledgment

We would like to express our sincere appreciation to Shenzhen Qianzhi Technology Co.,Ltd for their support in providing Lego toys manufacturing and defect samples.

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
