# Supplementary Materials for
# PAD: A Dataset and Benchmark for Pose-agnostic Anomaly Detection

**Qiang Zhou**[1][*]   **Weize Li**[1][*][†]   **Lihan Jiang**[2][†]   **Guoliang Wang**[1][†]
**Guyue Zhou**[1]   **Shanghang Zhang**[3]   **Hao Zhao**[1]
[1]AIR, Tsinghua University   [2]Wuhan University   [3]National Key Laboratory for Multimedia
Information Processing, School of Computer Science, Peking University
{bamboosdu, liweize0224}@gmail.com   shanghang@pku.edu.cn
zhaohao@air.tsinghua.edu.cn

## 1   MAD data collection pipeline

### 1.1   MAD-Simulated Data

By visiting the LEGO community, the author has obtained a number of open-source LEGO models. These models are constructed from parts provided by the Ldraw library (a basic LEGO parts library), exhibiting various small animal figures. To meet the requirements of the experiment, the author made fine adjustments and optimizations to the details of the models, such as edges and colors.During the data generation process, the author imported the necessary Ldraw parts into the Blender software and adjusted the angles and lighting of the models to achieve the best visual effects. To obtain more comprehensive 3D data, the author utilized a 360-degree surround camera technique to render the models from multiple angles.When setting up the cameras, the author used the circular surface centered on the vertex of the Z-axis as a reference, placing a camera every 15 degrees and adding cameras at the same intervals on the Z-axis. This setup allows multiple cameras to render simultaneously, thus obtaining richer and more comprehensive multi-angle model data.Sample parts and assembled products in Fig 1 and Fig 2.

### 1.2   MAD-Real data

The MAD-Real dataset is identical to MAD-Sim in terms of data structure and acquisition/division strategy for the training/test set, with the main differences being 1) acquisition from a real production line (with the sensitive background removed) introduces a relatively complex backgrounds and lighting environments as well as faint reflections on object surfaces. 2) The data size is smaller compared to MAD-Sim.

## 2   Lego family visualization with labels

Due to the large number of category labels within our MAD dataset, we provide the correspondence between category labels and images for easy visual identification and comparison, as shown in Fig.3.

---

[*]Equal contribution.
[†]Work done during internship at AIR.

37th Conference on Neural Information Processing Systems (NeurIPS 2023) Track on Datasets and Benchmarks.

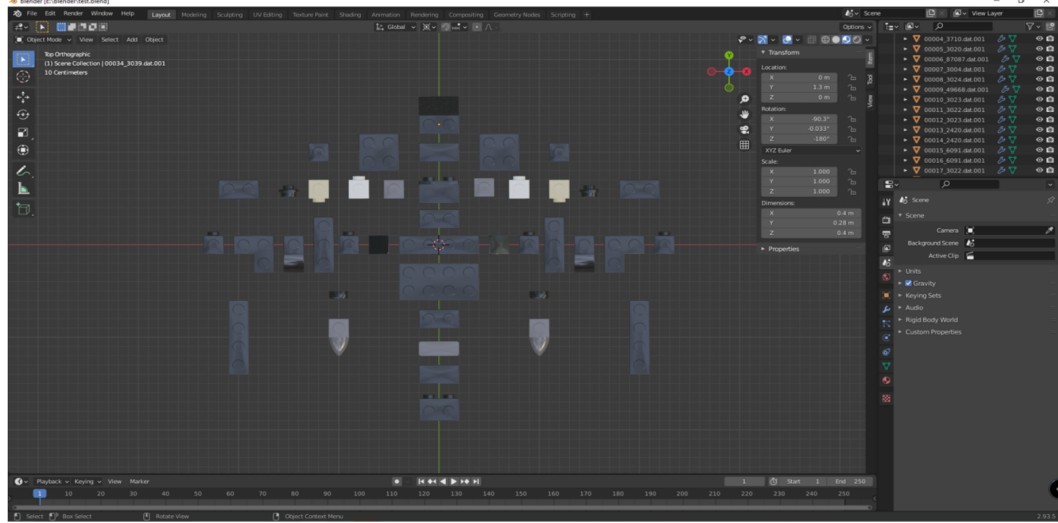

Figure 1: MAD-Simulated Data (Parts).

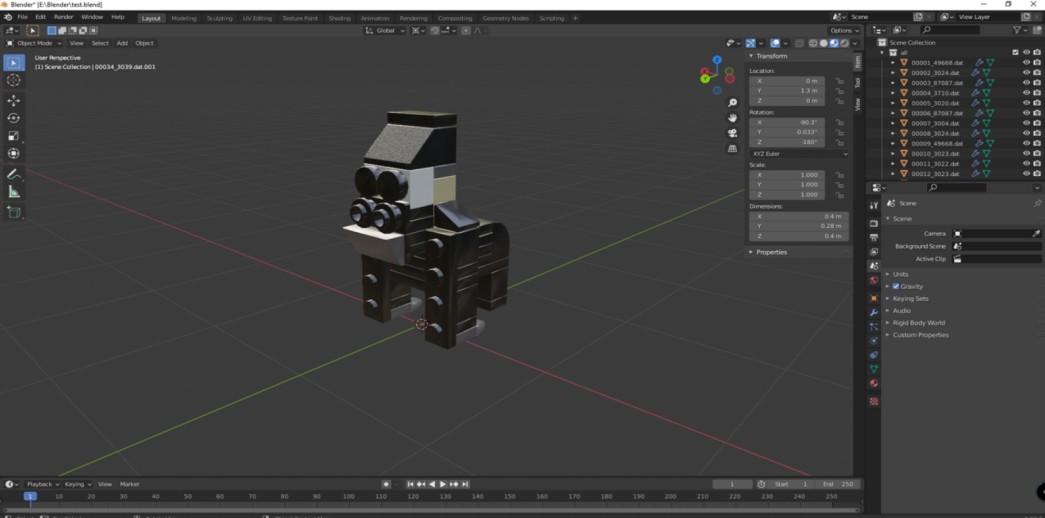

Figure 2: MAD-Simulated Data (Assembly).

## 3 Defect samples selection

When creating simulation data with defects, this study referred to several common types of defects on the LEGO production line, and selected 'Stains', 'Burrs', and 'Missing' as the main defect categories for the simulation data.

**"Burrs"** In the manufacturing process of LEGO toys, the occurrence of burrs as a defect can be attributed to several factors. Burrs are small, unwanted projections or rough edges that can form on the surface of LEGO bricks or components. Here are some reasons for the formation of burrs:

- Mold Design and Tooling: The design of the molds used to produce LEGO bricks plays a crucial role in determining the quality of the final product. If the mold design or tooling is not precise or if there are imperfections in the mold surface, it can lead to the formation of burrs during the injection molding process.

- Injection Molding Process: LEGO bricks are typically manufactured using injection molding, where molten plastic is injected into the mold cavities under high pressure. If the injection molding parameters such as temperature, pressure, or cooling time are not prop-

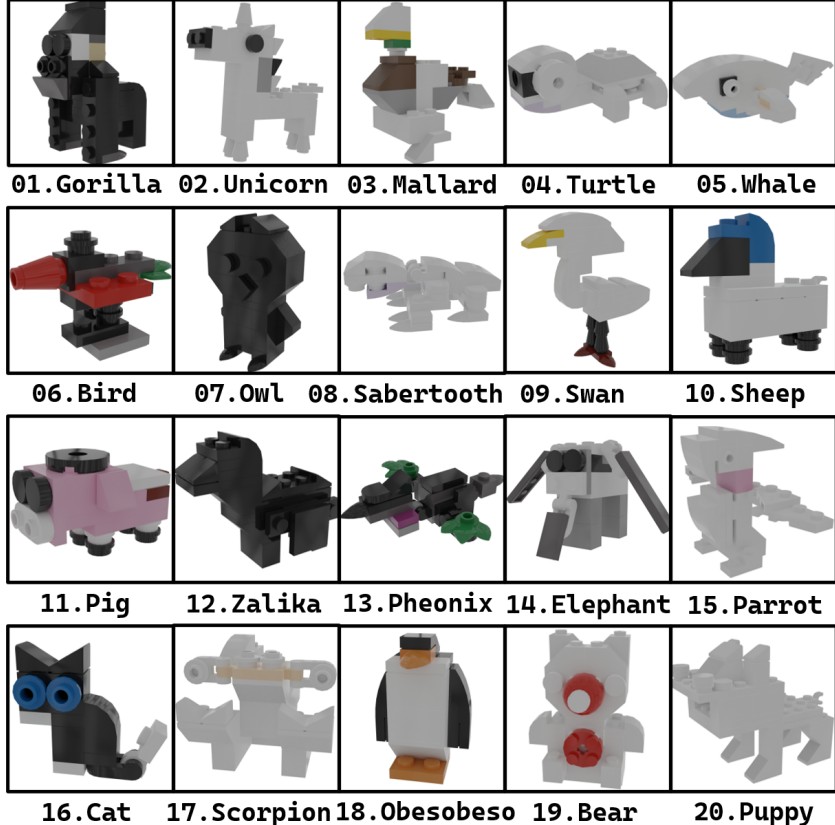

Figure 3: Category labels of MAD dataset.

erly controlled, it can result in uneven material flow or insufficient cooling, leading to the formation of burrs.

- Material Properties: The type of plastic used in LEGO bricks can also contribute to the formation of burrs. If the plastic material has a high viscosity or if it contains impurities, it can increase the likelihood of burr formation during the molding process.

- Mold Wear and Maintenance: Over time, the molds used in the production of LEGO bricks can wear out due to repeated use. As the mold surfaces degrade, they may become less smooth, resulting in the generation of burrs on the final product. Regular maintenance and proper upkeep of the molds are essential to minimize burr formation.

**"Stains"** In the manufacturing process of LEGO toys, the occurrence of stains as a defect can be attributed to various factors. Stains refer to discoloration or marks that appear on the surface of LEGO bricks or components. Here are some reasons for the formation of stains:

- Raw Material Quality: The quality of the plastic material used in LEGO production can affect the occurrence of stains. If the raw material contains impurities or is not properly formulated, it can result in discoloration or staining during the molding process.

- Mold Contamination: During the manufacturing process, molds used for LEGO bricks can sometimes get contaminated with foreign particles or residue from previous production runs. These contaminants can transfer to the surface of the bricks, leading to the formation of stains.

- Mold Release Agents: Mold release agents are substances applied to the molds to facilitate the easy removal of the molded bricks. If excessive or improper release agents are used, they can leave residues on the surface of the bricks, causing staining.

- Processing Conditions: The processing conditions during injection molding, such as temperature, pressure, and cycle time, can affect the occurrence of stains. If the conditions are not

properly controlled or if there are variations in the process parameters, it can lead to uneven material flow, incomplete filling of the molds, or other issues that contribute to stains.

**"Missing"** In the manufacturing process of LEGO toys, the occurrence of missing parts as a defect can be attributed to several reasons. Missing parts refer to situations where LEGO bricks or components are not included in the final packaged set as intended. Here are some possible causes of missing parts:

- Human Error: Mistakes can happen during the packaging process where human operators may accidentally overlook or omit certain LEGO parts while assembling the sets. This can occur due to manual counting errors or misjudgment during the packaging process.

- Quality Control Issues: Despite rigorous quality control measures, occasional errors in the inspection process can result in missing parts. If a defective batch is not identified during quality checks, it may proceed to packaging, leading to incomplete sets.

- Technical Malfunction: Mechanical or technical malfunctions in the assembly or packaging machinery can result in missing parts. For example, if there is a malfunction in the sorting or distribution system, it can lead to certain parts being skipped or not included in the final packaged sets.

- Supply Chain Issues: Disruptions or errors within the supply chain can also contribute to missing parts. This can include issues such as incorrect inventory management, misplacement during transportation or warehousing, or mishandling at any stage of the production and distribution process.

## 4 Attributes quantification

### 4.1 Shape Complexity

Each Lego brick in our dataset corresponds to a unique 3D shape. To better understand the impact of 3D shape complexity on anomaly detection, we intuitively use the number of triangular faces of the Lego toy model itself to represent the shape complexity. By measuring the 3D shape complexity, we can effectively evaluate the robustness of each anomaly detection algorithm under different shape complexities. For a fair comparison, we map the number of triangle slices to 0-100 for different classes of objects.

### 4.2 Texture Complexity

Similarly, object texture/color complexity is of equal concern. We intuitively use the computed color contrast to assess texture complexity: 1) Convert the image from the original color space to the Lab color space by dividing it into: L (luminance), a (green-red) and b (blue-yellow). 2) Extract the a and b channels from the Lab image, representing the variation between green and red (a channel) and blue and yellow (b channel). 3) Calculate the standard deviation of the a-channel values, representing the distribution or variation of color along the green-red axis. 4) Calculate the standard deviation of the b-channel values, indicating the spread or variation of the color along the blue-yellow axis. 5) Combine the standard deviations of the a and b channels to calculate the color contrast. A common method is to use the Euclidean distance formula, squaring the two standard deviations, summing them and taking the square root. The resulting value represents the color contrast of the image, the higher the value, the greater the color change, the more pronounced the chromatic aberration, i.e., the higher the complexity of the texture.

## 5 Additional evaluation for OmniposeAD

### 5.1 Implementation Details

**Backbone.** We use the EfficientNet-B4 pre-trained on Imagenet as our backbone, which is used in image retrieval and feature comparison. In practise, we choose the features from layer1 to layer5 of EfficientNet-B4 which have the channel of 24, 32, 56, 160, 448, respectively. And the feature from layer1 to layer5 are resized to the same size as the input image. Then the multi scale feature and the input image are concatenate together to form a feature map, f with the corresponding input pixel. For

our MAD dataset, the feature size are set as 224×224. Therefore, a feature map with the shape of 224×224×721 is obtained.

## 5.2 Anomaly localization visualization on the MAD-Sim dataset

To demonstrate the superiority of our proposed OmniposeAD on pose-agnostic setting, we provide visualizations of anomaly localization for all categories on the MAD-Sim dataset. Note that the first row denotes the input image, the second row denotes the ground truth, and the third row denotes the anomaly localization heat map.

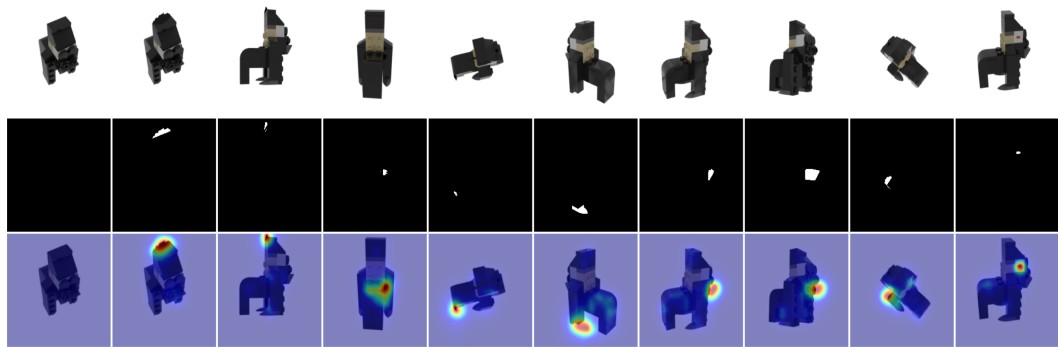

Figure 4: Anomaly detection results of Gorilla on MAD-Sim. From top to down: samples, ground-truth, and the anomaly score maps of OmniposeAD. The first column is the normal sample.

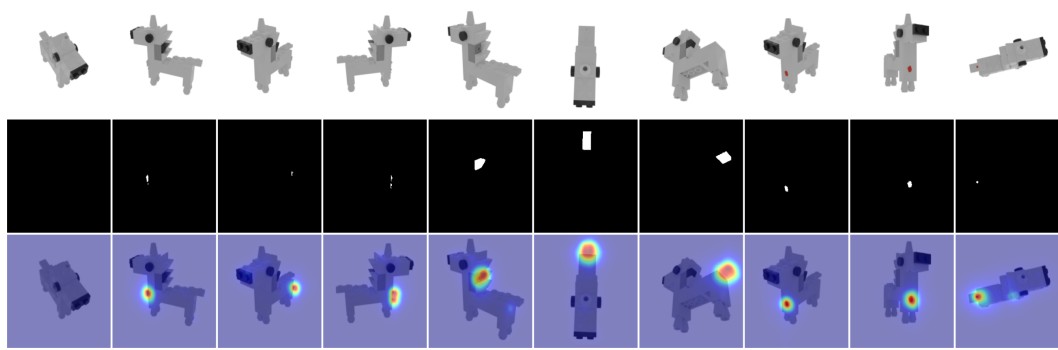

Figure 5: Anomaly detection results of Unicorn on MAD-Sim. From top to down: samples, ground-truth, and the anomaly score maps of OmniposeAD. The first column is the normal sample.

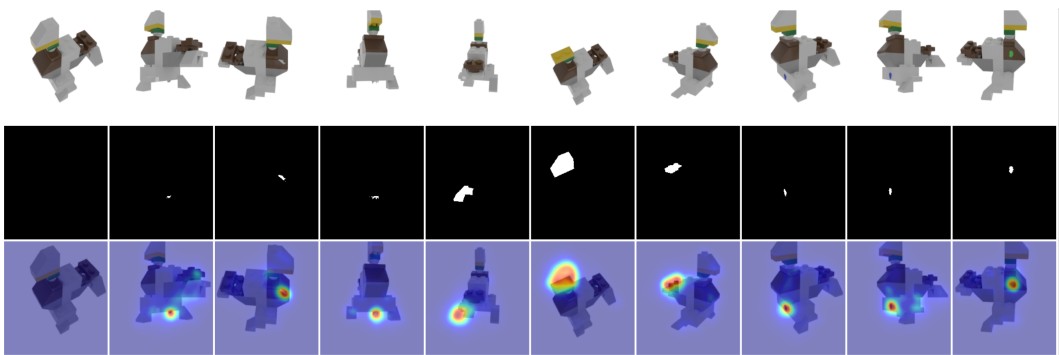

Figure 6: Anomaly detection results of Mallard on MAD-Sim. From top to down: samples, ground-truth, and the anomaly score maps of OmniposeAD. The first column is the normal sample.

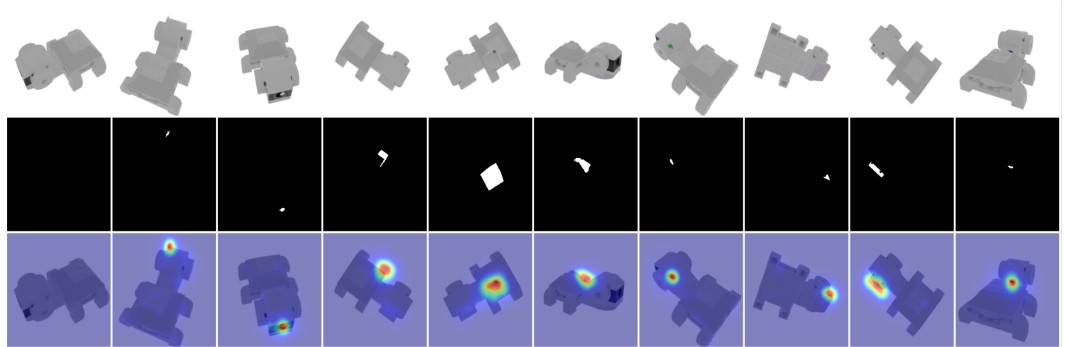

Figure 7: Anomaly detection results of Turtle on MAD-Sim. From top to down: samples, ground-truth, and the anomaly score maps of OmniposeAD. The first column is the normal sample.

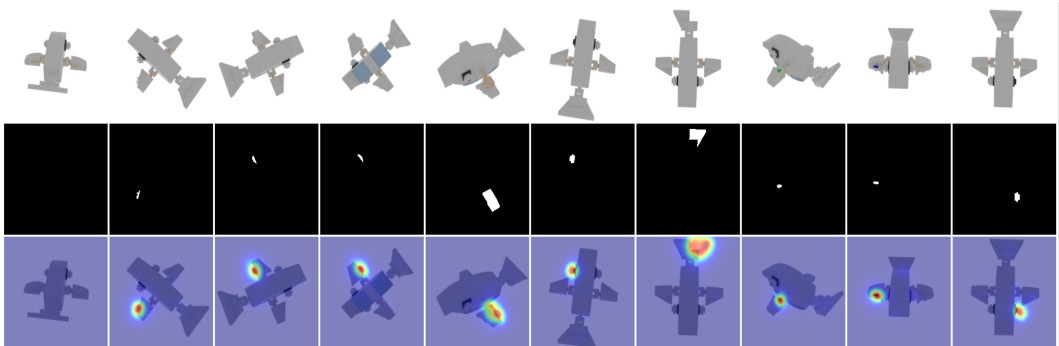

Figure 8: Anomaly detection results of Whale on MAD-Sim. From top to down: samples, ground-truth, and the anomaly score maps of OmniposeAD. The first column is the normal sample.

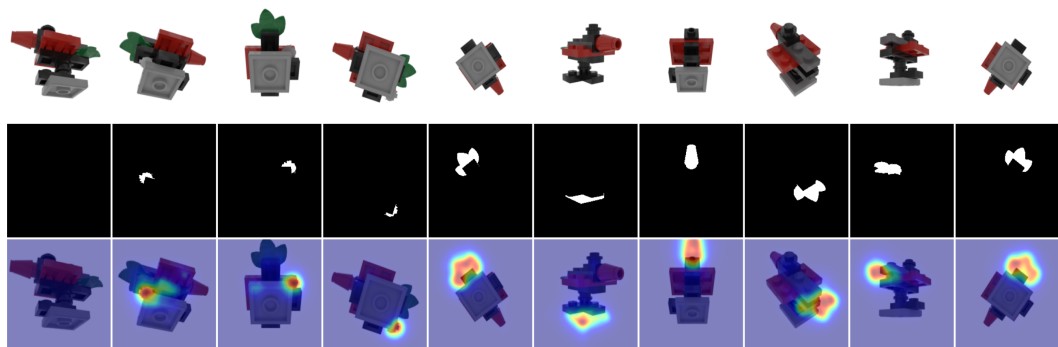

Figure 9: Anomaly detection results of Bird on MAD-Sim. From top to down: samples, ground-truth, and the anomaly score maps of OmniposeAD. The first column is the normal sample.

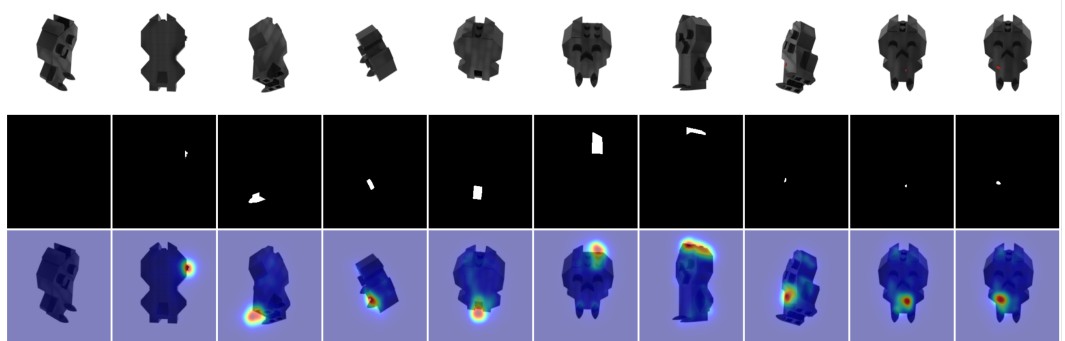

Figure 10: Anomaly detection results of Owl on MAD-Sim. From top to down: samples, ground-truth, and the anomaly score maps of OmniposeAD. The first column is the normal sample.

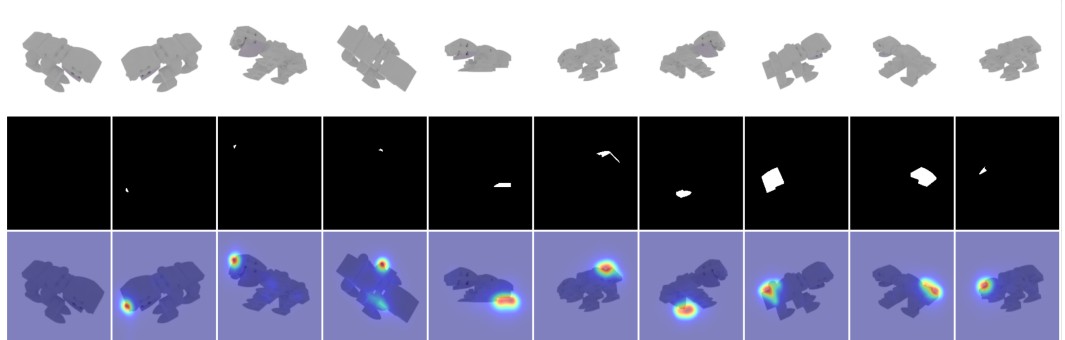

Figure 11: Anomaly detection results of Sabertooth on MAD-Sim. From top to down: samples, ground-truth, and the anomaly score maps of OmniposeAD. The first column is the normal sample.

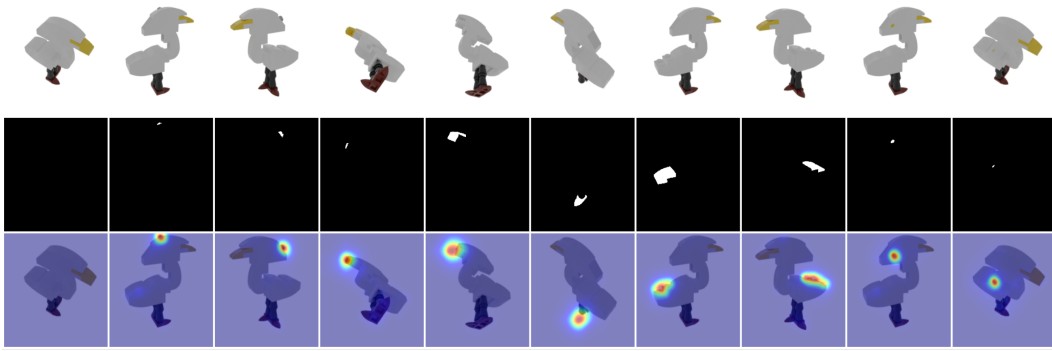

Figure 12: Anomaly detection results of Swan on MAD-Sim. From top to down: samples, ground-truth, and the anomaly score maps of OmniposeAD. The first column is the normal sample.

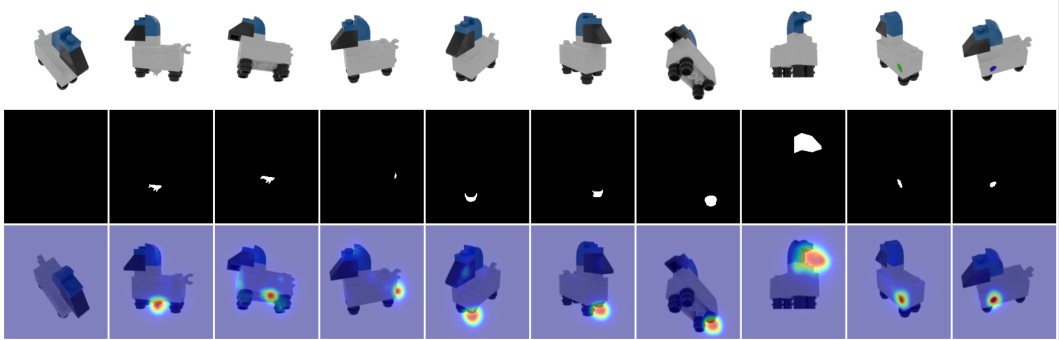

Figure 13: Anomaly detection results of Sheep on MAD-Sim. From top to down: samples, ground-truth, and the anomaly score maps of OmniposeAD. The first column is the normal sample.

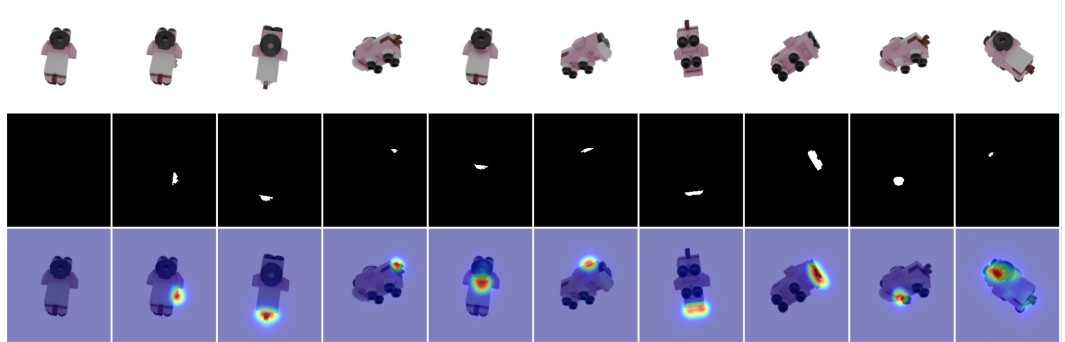

Figure 14: Anomaly detection results of Pig on MAD-Sim. From top to down: samples, ground-truth, and the anomaly score maps of OmniposeAD. The first column is the normal sample.

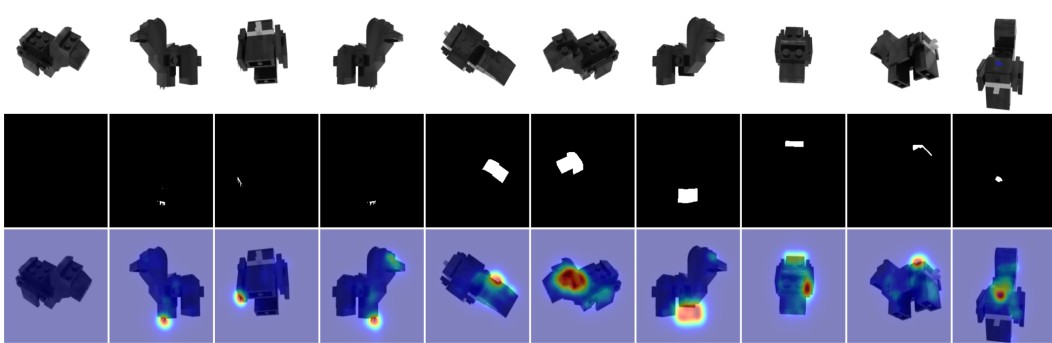

Figure 15: Anomaly detection results of Zalika on MAD-Sim. From top to down: samples, ground-truth, and the anomaly score maps of OmniposeAD. The first column is the normal sample.

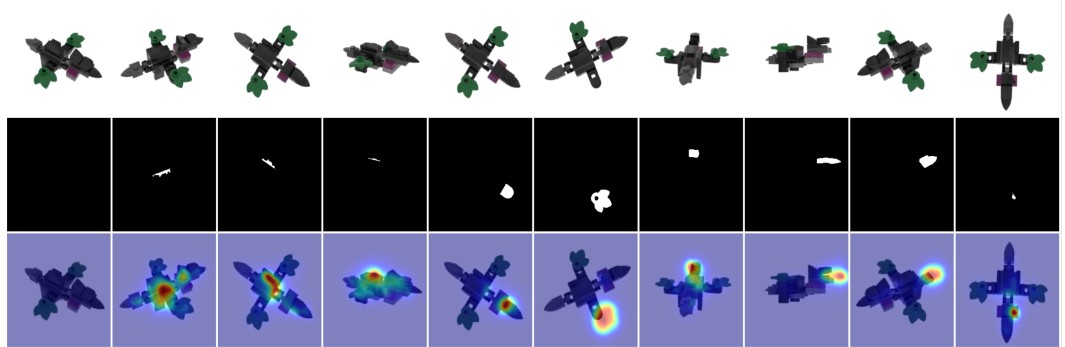

Figure 16: Anomaly detection results of Pheonix on MAD-Sim. From top to down: samples, ground-truth, and the anomaly score maps of OmniposeAD. The first column is the normal sample.

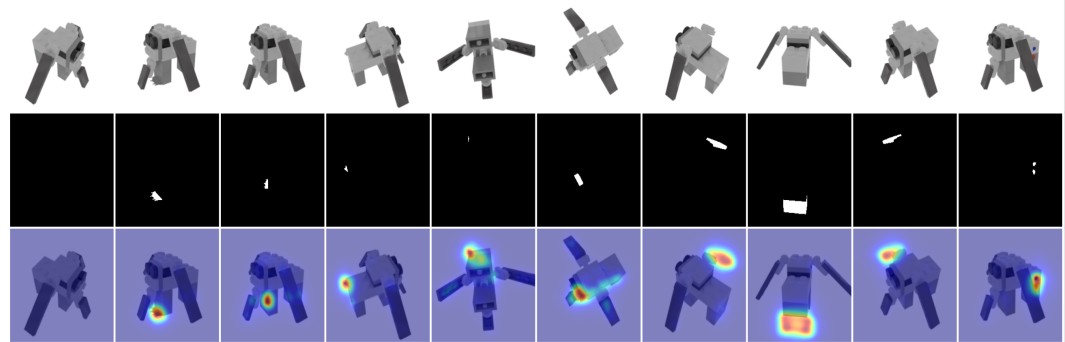

Figure 17: Anomaly detection results of Elephant on MAD-Sim. From top to down: samples, ground-truth, and the anomaly score maps of OmniposeAD. The first column is the normal sample.

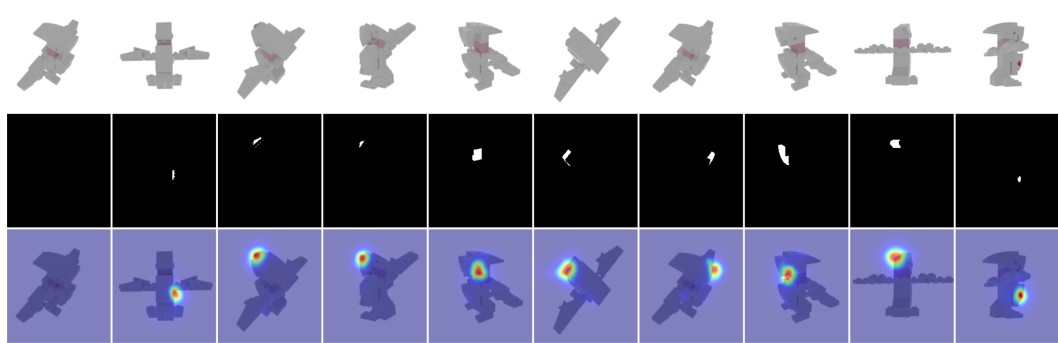

Figure 18: Anomaly detection results of Parrot on MAD-Sim. From top to down: samples, ground-truth, and the anomaly score maps of OmniposeAD. The first column is the normal sample.

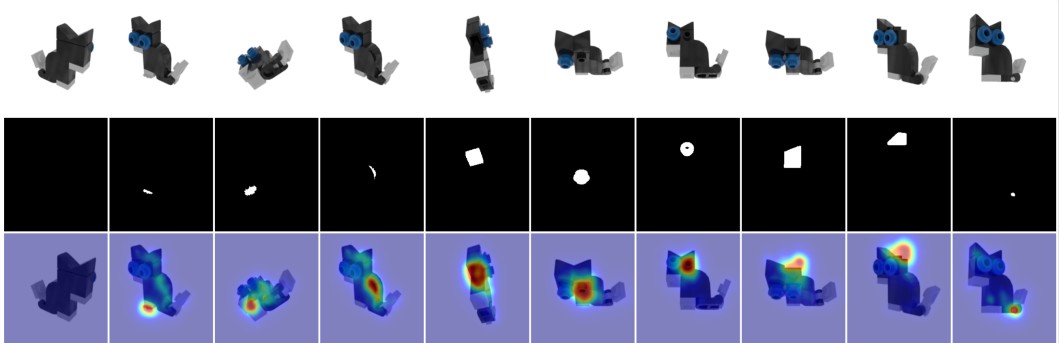

Figure 19: Anomaly detection results of Cat on MAD-Sim. From top to down: samples, ground-truth, and the anomaly score maps of OmniposeAD. The first column is the normal sample.

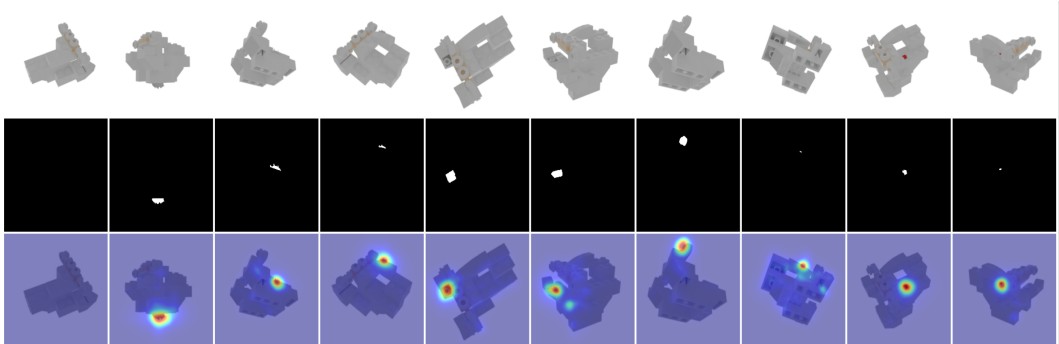

Figure 20: Anomaly detection results of Scorpion on MAD-Sim. From top to down: samples, ground-truth, and the anomaly score maps of OmniposeAD. The first column is the normal sample.

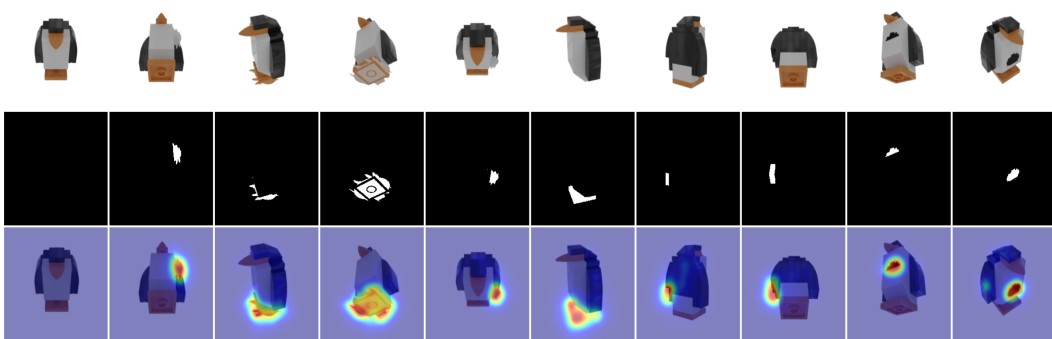

Figure 21: Anomaly detection results of Obesobeso on MAD-Sim. From top to down: samples, ground-truth, and the anomaly score maps of OmniposeAD. The first column is the normal sample.

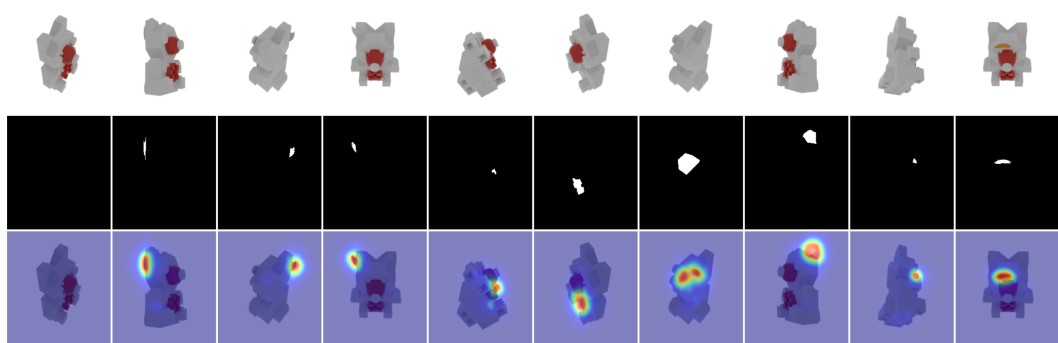

Figure 22: Anomaly detection results of Bear on MAD-Sim. From top to down: samples, ground-truth, and the anomaly score maps of OmniposeAD. The first column is the normal sample.

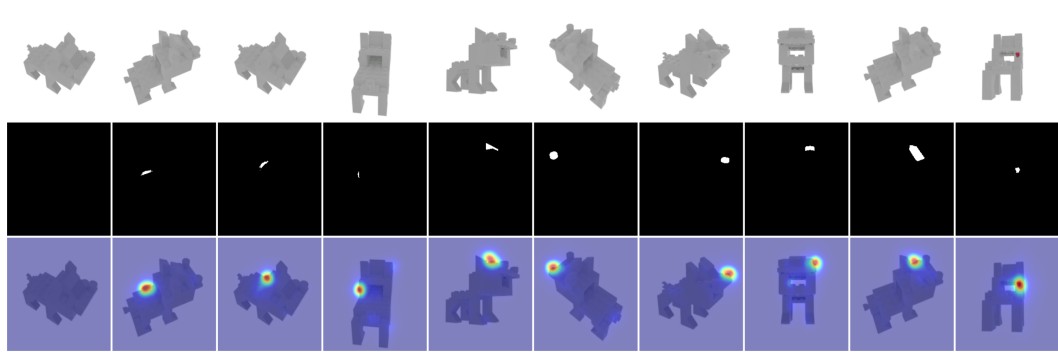

Figure 23: Anomaly detection results of Puppy on MAD-Sim. From top to down: samples, ground-truth, and the anomaly score maps of OmniposeAD. The first column is the normal sample.

Table 1: MAD-Sim result for several feature extractors. EfficientNet-b4 is the best results in both anomaly detection and location tasks. Each experiment keep the same hyperparameters, with 210 training images and 300 training epoches. We report mean for each experiment.

| | Pixel-AUROC | | | Image-AUROC | | |
|---|---|---|---|---|---|---|
| | Vanilla | VGG | EfficientNet | Vanilla | VGG | EfficientNet |
| Gorilla | 0.972 | 0.988 | 0.995 | 0.814 | 0.898 | 0.939 |
| Unicorn | 0.968 | 0.964 | 0.982 | 0.911 | 0.938 | 0.94 |
| Mallard | 0.970 | 0.973 | 0.974 | 0.862 | 0.848 | 0.847 |
| Turtle | 0.940 | 0.941 | 0.991 | 0.977 | 0.986 | 0.956 |
| Whale | 0.980 | 0.977 | 0.983 | 0.839 | 0.822 | 0.821 |
| Bird | 0.953 | 0.960 | 0.957 | 0.925 | 0.926 | 0.924 |
| Owl | 0.987 | 0.991 | 0.994 | 0.825 | 0.855 | 0.891 |
| Sabertooth | 0.975 | 0.973 | 0.985 | 0.973 | 0.969 | 0.961 |
| Swan | 0.976 | 0.979 | 0.989 | 0.876 | 0.870 | 0.865 |
| Sheep | 0.977 | 0.974 | 0.977 | 0.866 | 0.892 | 0.901 |
| Pig | 0.977 | 0.974 | 0.977 | 0.854 | 0.879 | 0.883 |
| Zalika | 0.985 | 0.987 | 0.991 | 0.890 | 0.847 | 0.886 |
| Pheonix | 0.991 | 0.992 | 0.994 | 0.855 | 0.899 | 0.858 |
| Elephant | 0.965 | 0.989 | 0.990 | 0.839 | 0.944 | 0.928 |
| Parrot | 0.990 | 0.988 | 0.995 | 0.972 | 0.978 | 0.974 |
| Cat | 0.974 | 0.973 | 0.977 | 0.891 | 0.870 | 0.849 |
| Scorpion | 0.950 | 0.951 | 0.959 | 0.935 | 0.933 | 0.915 |
| Obesobeso | 0.987 | 0.980 | 0.980 | 0.981 | 0.969 | 0.971 |
| Bear | 0.991 | 0.989 | 0.993 | 0.986 | 0.989 | 0.990 |
| Puppy | 0.978 | 0.974 | 0.988 | 0.915 | 0.935 | 0.934 |
| Mean | 0.974 | 0.976 | 0.984 | 0.899 | 0.912 | 0.912 |

## 5.3 Ablation Study

**Feature extractor selection.** In this experiment we study the performances of different ImageNet pre-trained model as feature extractors for our OmniposeAD framework. Results are listed in table 1.The Vanilla OmniposeAD just focuses on only pixel reconstruction. Besides, for VGG16 we select the 2nd, 3rd and 4th max pooling, which have the channel of 64, 32, 16, respectively. Here the feature size are set as 128×128, so we can obtain a feature map with the shape of 113. For both anomaly detection (Image AUROC) and location (Pixel AUROC) tasks, EfficientNet-B4 is still the best solution on average.

## 5.4 Segmentation Pixel-AUROC

After a lot of observations, we found that pixel AUROC was still high even though some anomaly detection results were not impressive. Therefore, this is not a good metric intuitively. We found that one of the reasons is that the anomalies are too small, and there exists too much background part. So we propose a new evaluation metric (segmentation pixel AUROC): extract the bounding box using segmentation from each image and then calculate the pixel AUROC. Here we compare our method with FAVAE, which looks significant in the evalution of pixel AUROC. In Table 2, we can demonstrate that our segmatnation pixel-AUROC are better than pixel AUROC.

Table 2: Both Pixel AUROC and Seg pixel AUROC are used to measure FAVAE and our method. After comparison, it is easy to find that the difference between the two methods is 0.088 while using Pixel-AUROC but 0.254 while using Segmentation pixel AUROC. This shows the superiority of our method and metric.

| | FAVAE | | Ours | |
|---|---|---|---|---|
| | Pixel-AUROC | Seg pixel AUROC | Pixel-AUROC | Seg pixel AUROC |
| Gorilla | 0.921 | 0.718 | 0.995 | 0.975 |
| Unicorn | 0.880 | 0.734 | 0.982 | 0.970 |
| Mallard | 0.853 | 0.675 | 0.974 | 0.950 |
| Turtle | 0.899 | 0.754 | 0.991 | 0.925 |
| Whale | 0.901 | 0.773 | 0.983 | 0.970 |
| Bird | 0.916 | 0.691 | 0.957 | 0.957 |
| Owl | 0.967 | 0.875 | 0.994 | 0.981 |
| Sabertooth | 0.945 | 0.832 | 0.985 | 0.951 |
| Swan | 0.874 | 0.678 | 0.989 | 0.973 |
| Sheep | 0.943 | 0.716 | 0.977 | 0.977 |
| Pig | 0.922 | 0.602 | 0.977 | 0.952 |
| Zalika | 0.864 | 0.669 | 0.991 | 0.992 |
| Pheonix | 0.924 | 0.841 | 0.994 | 0.980 |
| Elephant | 0.720 | 0.631 | 0.990 | 0.959 |
| Parrot | 0.877 | 0.756 | 0.995 | 0.982 |
| Cat | 0.940 | 0.648 | 0.977 | 0.975 |
| Scorpion | 0.884 | 0.732 | 0.959 | 0.931 |
| Obesobeso | 0.927 | 0.615 | 0.980 | 0.980 |
| Bear | 0.901 | 0.677 | 0.993 | 0.993 |
| Puppy | 0.856 | 0.654 | 0.988 | 0.988 |
| Mean | 0.896 | **0.714** | 0.984 | **0.968** |

# A   Boarder Impacts

Our proposed MAD dataset is dedicated to exploring the task of anomaly detection in a pose-agnostic setting. Instead of hard-to-access and expensive 3D information, such as point clouds, we can perform anomaly detection for any pose of an object using only 2D images from various viewpoints, which will revolutionize the practical application of object-centric anomaly detection.

# B   Data sheets for Dataset

## B.1   Motivation

1. **For what purpose was the dataset created?**
   See our Abstract.

2. **Who created the dataset (e.g., which team, research group) and on behalf of which entity (e.g., company, institution, organization)?**
   The authors listed on this paper, which include students and researchers from AIR, Tsinghua University, Wuhan University and Peking University.

3. **Who funded the creation of the dataset?**
   AIR, Tsinghua University.

### B.2 Composition

1. **What do the instances that comprise the dataset represent (e.g., documents, photos, people, countries)?**
   The representative examples that make up the dataset are animal images assembled by Lego, such as chimpanzees and unicorns, and contain both simulation and real data. We provide training data and test data with defective samples for each type of data.

2. **How many instances are there in total(of each type,if appropriate)?**
   The dataset contains a total of 20 types of Lego, with approximately 550 images of each type.

3. **Does the dataset contain all possible instances or is it a sample (not necessarily random) of instances from a larger set?**
   No.

4. **What data does each instance consist of?**
   We provide raw data and flawed data for each type of instance data, including simulation data and real data.

5. **Is there a label or target associated with each instance?**
   We provide label information for each type of instance, such as "Good", "Stains", "Burrs", and "Missing".

6. **Is any information missing from individual instances?**
   No.

7. **Are relationships between individual instances made explicit (e.g., users' movie ratings, social network links)?**
   No.

8. **Are there recommended data splits (e.g., training, development/validation, testing)?**
   Referring to the splitting mode of existing anomaly detection data, divide the data without anomalies into training data. In the test data, add various types of anomaly data and anomaly-free data, and place them in folders with different labels.

9. **Are there any errors, sources of noise, or redundancies in the dataset?**
   No.

10. **Is the dataset self-contained, or does it link to or otherwise rely on external resources (e.g., websites, tweets, other datasets)?**
    The dataset is independent, and we will provide the official version of the dataset.

11. **Does the dataset contain data that might be considered confidential (e.g., data that is protected by legal privilege or by doctor–patient confidentiality, data that includes the content of individuals' non-public communications)?**
    No.

12. **Does the dataset contain data that, if viewed directly, might be offensive, insulting, threatening, or might otherwise cause anxiety?**
    No.

### B.3 Collection Process

1. **How was the data associated with each instance acquired?**
   The data is directly obtainable.

2. **What mechanisms or procedures were used to collect the data (e.g., hardware apparatuses or sensors, manual human curation, software programs, software APIs)?**
   Use simulation software Blender to render and generate simulation data, and use camera to collect real data.

3. **If the dataset is a sample from a larger set, what was the sampling strategy (e.g., deterministic, probabilistic with specific sampling probabilities)?**
   Not applicable.

4. **Who was involved in the data collection process (e.g., students, crowdworkers, contractors)?**
   Researchers and students in AIR.

5. **Over what timeframe was the data collected?**
   The data was collected from September 2022 to May 2023.

6. **Were any ethical review processes conducted (e.g., by an institutional review board)?**
   No.

## B.4 Preprocessing/cleaning/labeling

1. **Was any preprocessing/cleaning/labeling of the data done (e.g., discretization or bucketing, tokenization, part-of-speech tagging, SIFT feature extraction, removal of instances, processing of missing values)?**
   Yes, see our main paper.

2. **Was the "raw" data saved in addition to the preprocessed/cleaned/labeled data (e.g., to support unanticipated future uses)?**
   No.

3. **Is the software that was used to preprocess/clean/label the data available?**
   Yes, Blender and Photoshop is available for anyone.

## B.5 Uses

1. **Has the dataset been used for any tasks already?**
   Yes, we include object anomaly detection and localization tasks in our main paper.

2. **Is there a repository that links to any or all papers or systems that use the dataset?**
   Yes, see repository https://github.com/EricLee0224/PAD

3. **What (other) tasks could the dataset be used for?**
   Visual anomaly detection and localization tasks under the PAD setting.

4. Is there anything about the composition of the dataset or the way it was collected and preprocessed/cleaned/labeled that might impact future uses?
   No.

5. **Are there tasks for which the dataset should not be used?**
   No.

## B.6 Distribution

1. **How will the dataset will be distributed (e.g., tarball on website, API, GitHub)? Does the dataset have a digital object identifier (DOI)?**
   Yes. It will be publicly available for download.

2. **When will the dataset be distributed?**
   See repository https://github.com/EricLee0224/PAD

3. **Will the dataset be distributed under a copyright or other intellectual property (IP) license, and/or under applicable terms of use (ToU)?**
   We licensed our MAD dataset under CC BY-NC-SA 4.0. Copyright of the original LEGO and models are owned by their creators respectively.

4. **Have any third parties imposed IP-based or other restrictions on the data associated with the instances?**
   No, the resources for all instances are open source.

5. **Do any export controls or other regulatory restrictions apply to the dataset or to individual instances?**
   No.

## B.7 Maintenance

1. **Who will be supporting/hosting/maintaining the dataset?**
   The authors of paper will be maintaining the dataset.

2. **How can the owner/curator/manager of the dataset be contacted (e.g., email address)?**
   Please contact: bamboosdu@gmail.com or liweize0224@gmail.com; For OmniposeAD, please contact: mr.lhjiang@gmail.com.

3. **Is there an erratum?**
   there have any errors to be pointed out, we will provide the erratum which you can found in our Github repository.

4. **If the dataset relates to people, are there applicable limits on the retention of the data associated with the instances (e.g., were the individuals in question told that their data would be retained for a fixed period of time and then deleted)?**
   Not applicable.

5. **Will older versions of the dataset continue to be supported/hosted/maintained?**
   Yes, all the dataset no matter older version or newest version which will be maintaing on Github repository.

6. **If others want to extend/augment/build on/contribute to the dataset, is there a mechanism for them to do so?**
   We provide CC BY-NC-SA 4.0 license for our dataset, so others can freely contribute in our dataset.