# OpenReview forum: "PAD: A Dataset and Benchmark for Pose-agnostic Anomaly Detection"
_NeurIPS.cc/2023/Track/Datasets_and_Benchmarks — NeurIPS 2023 Datasets and Benchmarks Poster_

### Official Review · Reviewer_upSJ · 2023-07-05
**A dataset for anomaly detection that poses an interesting new problem, but there are some issues**

**Rating:** 6
**Confidence:** 4

**Strengths:**

The problem setting of pose-agnostic anomaly detection is a valuable new addition to the field of anomaly detection. While the single-pose setting of other visual anomaly detection datasets is appropriate for certain applications (e.g., in very controlled industrial settings), the pose-agnostic setting will be more appropriate for some other applications.
The variety introduced by different poses can make the problem more challenging.

The benchmark conducted by the authors shows that many recently introduced visual anomaly detection algorithms are not very suited to the pose-agnostic setting, indicating the need for new approaches. The introduced NeRF-based method is a valuable first step in that direction.

**Additional Feedback:**

In Section 5.5 the authors write that the results in Table 3 demonstrate "only marginal degradation in performance as the amount of training data decreases". In my opinion, some of the differences are more than marginal decreases, for example 0.931 to 0.803 image-level AUROC for the object "Puppy".

In Table 2, the bolded numbers are not always the highest ones in the respective category. For example, the pixel score for "bird" is higher for OCRGAN than for "Ours", but the latter is bolded. These issues should be fixed for the final submission. (This does not change the general conclusions that can be drawn from the table.)

I found a few typos:
- in the legend of Table 1: "comples" -> "complex"?
- in the legend of Figure 5: "performence" -> "performance"?
- in line 282 and 289: "pose diagnostic" -> "pose agnostic"?
- Should the object category "pheonix" be named "phoenix"?

The authors might want to check if there are any further typos before submitting the final version of the paper.

**Clarity:**

In general, the paper is well-written and easy to understand.

One point where I would appreciate a clearer description is in the construction of the train and test sets. Section 3.1 states that "To generate training data with more normal instances, we used the same approach but with slightly modified parameters.". Does this mean that all training images were constructed with different parameters than the test images? Or were these different parameters used to produce additional training data? A more detailed explanation of the differences between the (anomaly-free) test data and the train data would be helpful to assess whether the dataset splits were constructed in a sound way. It is important that test images are not just repetitions of training samples, while also not introducing an unintended dataset shift (except for the anomalous samples, of course).

**Correctness:**

The simulated part of the dataset appears to be constructed in a sound way. The constuction of the train and test sets could use a more detailed description (cf. my comment in "Clarity" below) in order to assess whether there are any problems in this regard.

I did not find any problems with the benchmark of the various anomaly detection methods.

I think formula (5) on page 7 is incorrected. It should be $AUROC = \int TPR\ dFPR$.

The authors study the correlation between the color contrast of objects and the performance of methods. I am not sure whether the definition of color constrast in the supplementary material is a sound approach. It seems to be more of an ad-hoc definition. The reasons for choosing this definition are unclear. Is this a common approach in the literature? If that is the case, a literature reference would be helpful.

In Section 4 of the supplementary material, the authors introduce the "Segmentation Pixel-AUROC" metric. If I understand it correctly, this metric is computed by calculating the AUROC only inside the bounding boxes of the regions segmented by an anomaly detection algorithm. This seems problematic to me as it would ignore false negatives outside of the bounding boxes. Additionally, this would mean that the pixels used for computing the metric would differ from method to method. (If the bounding boxes are based on the ground truth regions instead, false positive anomaly detections outside of the bounding boxes would be ignored.)

**Documentation:**

The simulated part of the dataset is sufficiently documented. However, there is not a lot of information about the real-world images in the dataset. At the time of writing this review, the latter part of the dataset was not accessible, so it was not possible to assess whether there are any problems with that part of the dataset, making a thorough review impossible.

In section D of in the supplementary material, two questions were not answered: Question 3 in D.5 and Question 4 in D.6.

The authors mention implementation and training details, though not for all evaluated methods. They write that code will be publicly available in a GitHub repository. At the time of writing this review, the linked repository did not yet contain the all necessary information. If the missing parts are added upon publication, reproducing the results should be possible.

**Ethics:**

I do not have any concerns that would require an ethics review.

**Limitations:**

While the authors mention a limitation of their proposed baseline method, limitations of the dataset (cf. my comments in "Opportunities For Improvement") are not sufficiently discussed in the paper.

**Opportunities For Improvement:**

The simulated images show the objects in front of a homogeneous white background. This is an artificial setting that differs from what one would expect from a real-world setting. Any complications introduced by a background are avoided. Good performance of a method in the artificial setting might not necessarily transfor to a more realistic setting.

While the LEGO animals differ significantly in their shape, they still belong to essentially the same domain. They are rigid objects and do not contain different kinds of textures. A larger diversity in the dataset (e.g., objects with natural variations) would be a valuable improvement.

**Relation To Prior Work:**

Related work is discussed in sufficient detail and the differences to the proposed dataset and baseline method are clearly stated.

The authors might consider also mentioning the VisA dataset (introduced in Yang Zou et al., "SPot-the-Difference Self-supervised Pre-training for Anomaly Detection and Segmentation", ECCV 2022, https://link.springer.com/chapter/10.1007/978-3-031-20056-4_23) in the "Related Work" section as that dataset is similar to some of the others mentioned there.

**Summary And Contributions:**

The authors introduce a new dataset ("Multi-pose Anomaly Detection (MAD) dataset") for anomaly detection in images. The objects in the images can appear in any pose. This distinguishes the dataset from other anomaly detection datasets where the objects usually appear in a (roughly) fixed pose. The authors call this problem setting "pose-agnostic anomaly detection".
The dataset contains more than 10000 images and is divided into 20 object categories. The objects are constructed from LEGO bricks and have shapes that resemble different animals.
The dataset contains simulated as well as real images of the objects.
There are three kinds of anomalies present in the dataset: burrs, stains and missing parts. Pixel-precise ground truth annotations of the anomalies are provided in the dataset.

The authors introduce a new baseline method for the proposed task of pose-agnostic anomaly detection. This approach is based on neural radiance fields (NeRFs). The pipeline consists roughly of the following steps:
1. Estimating the object pose of the object in an images.
2. Using a NeRF trained on the anomaly-free data to render the object in that pose.
3. Compare the input image and the rendered image using a pretrained feature extractor. Differences indicate the presence of an anomaly.

The authors conduct a benchmark of their proposed method and ten other anomaly detections on the MAD dataset. The results show that the proposed NeRF-based method significantly outperforms the other approaches (which were not designed specifically for pose-agnostic object detection).
Additionally, the authors study the correlation between object attributes (shape complexity and color contrast) and method performance.

---

> ### Author Response · Authors · 2023-08-21
> **Response to Reviewer upSJ (1 of 5).**
>
> We express our sincere gratitude to Reviewer for conducting a meticulous review of our manuscript and providing extensive and valuable feedback. Your comprehensive comments have greatly contributed to identifying areas for improvement. We have carefully evaluated your feedback and present our responses below:
>
> **Q1:** Simulated Images and Realistic Setting
>
> > "The simulated images show the objects in front of a homogeneous white background. This is an artificial setting that differs from what one would expect from a real-world setting. Any complications introduced by a background are avoided. Good performance of a method in the artificial setting might not necessarily transfer to a more realistic setting."
>
> **A1:** We appreciate your concerns about background settings in the MAD-Sim dataset. The considerations we took into account before placing objects on a homogeneous white background in the MAD-Sim dataset were:
>
> 1) The unified white background allows us to focus on evaluating the performance of different AD algorithms for object pose-agnostic anomaly detection, i.e., focusing on the ability to capture object-centric anomaly-free information over the full range of viewing angles;
>
> 2) We referred to the experience of background settings in previous datasets, such as the MvTec-AD dataset where the background settings of objects are black or white.
>
> In fact, we realized at the beginning that this ideal artificial setting does not match the real-world setting, which is much more intricate and complex and is characterized by varying factors such as background and lighting conditions. For this reason, we sampled 20 types of objects in reality from multiple angles and selected complex backgrounds and natural lighting conditions to form the MAD-Real dataset. Currently, our MAD-Real dataset is under open source review. As soon as the review is complete, we will make it available to the public.
>
> Nevertheless, we remain committed to giving reviewers as much information as possible for further evaluation. In order to demonstrate that our method (OmniposeAD) performs well in artificial environments as well as being able to work in real environments, we conducted additional experiments: We selected several LEGO toys and sampled and tested them in real-world environments (in-the-wild scene), noting that these scenes are more complex than the backgrounds in the MAD-Real dataset. A visualization of the test results can be found in our repository: https://github.com/EricLee0224/PAD#36-in-the-wild-ad-omniposead-results
>
> **Q2:** Diversity in LEGO Animal Dataset
>
> > "While the LEGO animals differ significantly in their shape, they still belong to essentially the same domain. They are rigid objects and do not contain different kinds of textures. A larger diversity in the dataset (e.g., objects with natural variations) would be a valuable improvement."
>
> **A2:** Our work aims to explore the first steps in the performance of AD algorithms under challenging PAD settings. Lego animal toys with diverse shapes were chosen to compose the MAD dataset and, for the first time, multi-pose observation samples and pose information were introduced. In follow-up work we will try to introduce objects from multiple domains to further explore the PAD setting.
>
> Since this study is based on a real industrial production project, through our research, we consider that object anomaly detection is usually oriented towards industrial manufacturing applications, where the objects to be detected are usually rigid standardized parts. That's why we chose rigid LEGO animal toys. In fact, the individual constituent parts on the 20 Legos are solid colors with no color texture variations, but their surface tactile textures contain smooth and frosted. However, we have observed that it is difficult to distinguish between different tactile texture variations, both on simulation-rendered RGB images and on real RGB pictures taken.
>
> We agree with the reviewers that defining anomalies in naturally varying objects is important in the field of object anomaly detection, and we have listed this as one of the limitations of the MAD dataset that we hope to address in future research.

---

> > ### Author Response · Authors · 2023-08-21
> > **Response to Reviewer upSJ (2 of 5).**
> >
> > **Q3:** Addressing Dataset Limitations
> >
> > > "While the authors mention a limitation of their proposed baseline method, limitations of the dataset (cf. my comments in "Opportunities For Improvement") are not sufficiently discussed in the paper."
> >
> > **A3:** The reviewer's point about discussing the limitations of the dataset is well taken. However, due to page limitations, we briefly mentioned potential limitations in the supplementary material (submission version), and in response to the reviewer's comments we have added the limitations of the work in detail in Sec.6 (revised version), including a discussion of the limitations of the dataset:
> >
> > "For the MAD dataset we developed as a starting point for exploring PAD, only rigid objects and not naturally varying objects were included, and the complexity and diversity of industrial products made it difficult to collect and quantify a complete sample of the diversity of characteristics. "
> >
> > We look forward to discussing with the reviewers additional dataset limitations that warrant inclusion.
> >
> > **Q4:** Construction of Dataset Splits
> >
> > > "The simulated part of the dataset appears to be constructed in a sound way. The construction of the train and test sets could use a more  detailed description (cf. my comment in "Clarity" below) in order to  assess whether there are any problems in this regard."
> >
> > **A4:** For this issue, we have merged the answers, as detailed in A8.
> >
> > **Q5:** Formula (5) for AUROC Definition
> >
> > > "I think formula (5) on page 7 is incorrected. It should be: "
> >
> > $$AUROC = \\int(TPR) dFPR$$
> >
> > **A5:** We thank the reviewers for their comments regarding the accuracy of the AUROC definition formula (5). We have thoroughly reviewed and amended the formula to ensure its correctness.
> >
> > **Q6:** Definition of Color Contrast Metric
> >
> > > "The authors study the correlation between the color contrast of objects and the performance of methods. I am not sure whether the definition of color contrast in the supplementary material is a sound approach. It  seems to be more of an ad-hoc definition. The reasons for choosing this  definition are unclear. Is this a common approach in the literature? If  that is the case, a literature reference would be helpful."
> >
> > **A6:** We appreciate the reviewer's feedback and would like to address the concerns raised regarding the color contrast and its selection in our study.
> >
> > The choice of color contrast definition was indeed made with careful consideration of its relevance to real-world anomaly detection scenarios and its intuitive interpretation: The rationale behind introducing color contrast as a quantifiable attribute in our research is rooted in the desire to enhance the practical applicability of anomaly detection algorithms in real-world deployment scenarios (Refer to revised version Sec3.3).
> >
> > When encountering an object for anomaly detection, observers often rely on their immediate perception of the object's color attributes. We believe that considering color contrast as quantifiable attributes for evaluating anomaly detection methods aligns well with the natural way people perceive and assess objects for potential anomalies.
> >
> > The Supplementary Material describes the details of computational steps we chose to use to quantify color contrast, rather than a ad-hoc definition (definitions are usually fixed, but there are a variety of ways to compute contrast.)
> >
> > In response to the issues raised by the reviewers, we have added some references that guided our choice of color contrast as one of the object attributes:
> >
> > - Forsyth, D. A., & Ponce, J. (2012). "Computer Vision: A Modern Approach." In this textbook, the chapter on color covers various aspects of color perception and representation, which could provide a basis for discussing color contrast.
> > - Ebner, M. (2007). "Color Constancy." Wiley Encyclopedia of Computer Science and Engineering. This article discusses color constancy, which relates to color perception under varying lighting conditions, and could offer insights into color contrast discussions.
> > - Khan, Fahad Shahbaz, et al. "Color attributes for object detection." 2012 IEEE Conference on Computer Vision and Pattern Recognition.
> > - Fan, Guofan, et al. "Point-GCC: Universal Self-supervised 3D Scene Pre-training via Geometry-Color Contrast." *arXiv preprint *(2023).

---

> > > ### Author Response · Authors · 2023-08-21
> > > **Response to Reviewer upSJ (3 of 5).**
> > >
> > > **Q7:** Segmentation Pixel-AUROC Metric
> > >
> > > > "In Section 4 of the supplementary material, the authors introduce the "Segmentation Pixel-AUROC" metric. If I understand it correctly, this  metric is computed by calculating the AUROC only inside the bounding boxes of the regions segmented by an anomaly detection algorithm. This seems problematic to me as it would ignore false negatives outside of  the bounding boxes. Additionally, this would mean that the pixels used  for computing the metric would differ from method to method. (If the  bounding boxes are based on the ground truth regions instead, false  positive anomaly detections outside of the bounding boxes would be ignored.)"
> > >
> > > **A7:** We would like to draw attention to the fact that the metric we used in our benchmarking experiments was the standard pixel/image-level AUROC, and **did not use the "Segmentation pixel-AUROC"** from the Supplementary Material.
> > >
> > > The reason we chose to put "Segmentation pixel-AUROC" in the Supplementary Material is:
> > >
> > > "Both the reconstruction-based method and our proposed OmniposeAD essentially reconstruct the normal reference. In the quantitative results of the anomaly localization task, the representative reconstruction-based method FAVAE achieved 89.4, and we further visualized the reconstructed references to explore the gap. Figure.\ref{fig6} shows that FAVAE does not reconstruct references for unseen poses well (the quantitative results do not align with visual facts), hinting that the existing evaluation metric are potentially risky in the PAD setting. Our further discussion in Sec.4 of the Supplementary Material." (Revised version Sec.5.3)
> > >
> > > The idea of **"Segmentation pixel-AUROC"** may be premature (as mentioned in the comments), but we wanted to provoke more thought. But we still value the comments made by reviewers and are happy to have more discussion:
> > >
> > > In benchmarking, we found that background in the object anomaly detection task does have a significant impact on performance (this is true on most AD datasets, including MAD-Real and in-the-wild). However, with the rapid development of image segmentation tools, especially SAM and SegGPT, it is easy to segment the object from the background (note that it is not to segment semantic-level anomalies, but only objects), which makes object-centric anomaly detection possible. So we try to use Segmentation pixel-AUROC as a new metric, i.e., we try to segment the object into regions of interest, and then only compute the performance inside the bnd box.
> > >
> > > We agree with the reviewers' insights. We think that for the ignored false negatives mentioned by the reviewer, they usually occur in texture anomaly detection and not in object anomaly detection (because we only label GT on objects and do not consider the background).
> > >
> > > **Q8:** Clarity in Dataset Construction
> > >
> > > > "One point where I would appreciate a clearer description is in the construction of the train and test sets. Section 3.1 states that "To  generate training data with more normal instances, we used the same  approach but with slightly modified parameters.". Does this mean that  all training images were constructed with different parameters than the test images? Or were these different parameters used to produce additional training data? A more detailed explanation of the differences  between the (anomaly-free) test data and the train data would be helpful to assess whether the dataset splits were constructed in a sound way. It is important that test images are not just repetitions of training samples, while also not introducing an unintended dataset shift  (except for the anomalous samples, of course)."
> > >
> > > **A8:** We recognize the importance of clearly describing the dataset construction process and provide detailed explanations for any misunderstandings caused by unclear descriptions in the paper. To this end, we provide a comprehensive explanation that emphasizes the differences between the training and test data and ensures that the dataset splits are justified.
> > >
> > > In fact, we have considered at the beginning of the construction of the dataset to try to avoid duplications between the test set and the training set at any level within the same domain.
> > >
> > > 1) the parameters of the training set and the test set differ only in the pose information (we would like to introduce in the test set some pose samples that have not been seen in the training set in order to validate the PAD settings)
> > > 2) the presence of both normal and anomalous samples in the test set (to validate image-level performance).
> > >
> > > In addition, we have removed the sentence that created the misunderstanding in the revised version.

---

> > > > ### Author Response · Authors · 2023-08-21
> > > > **Response to Reviewer upSJ (4 of 5).**
> > > >
> > > > **Q9:** Mentioning VisA Dataset in Related Work
> > > >
> > > > > "The authors might consider also mentioning the VisA dataset (introduced  in Yang Zou et al., "SPot-the-Difference Self-supervised Pre-training  for Anomaly Detection and Segmentation", ECCV 2022) in the "Related Work" section as that dataset is similar to some of the others mentioned there."
> > > >
> > > > **A9:** We thank the reviewer for bringing the VisA dataset to our attention! We have included discussion and description for VisA dataset in "Related Work"(line88-90) and Table.1, respectively. Please see details in revised version.
> > > >
> > > > **Q10:** Description of MAD-Real Dataset
> > > >
> > > > > "There is not a lot of information about the real-world images in the dataset. At the time of writing this review, the latter part of  the dataset was not accessible, so it was not possible to assess whether there are any problems with that part of the dataset, making a thorough review impossible."
> > > >
> > > > **A10:** The MAD-Real dataset is still under open source review, and as soon as it completes the review and is allowed to be made public, we will update the download link on Github repo.
> > > >
> > > > I would be happy to provide you with more information about the MAD-Real dataset for you to evaluate if there are any issues with this part of the dataset. The MAD-Real dataset is identical to MAD-Sim in terms of data structure and split strategy for the training/test set, with the main differences being:
> > > >
> > > > 1) The data collected from a real production line (note that the background has been removed from the part of the production line involving sensitive information) reflects a complex background and lighting environment, as well as weak reflections on the surface of the object.
> > > > 2) Compared to MAD-Sim, the amount of data is smaller, which is consistent with the actual situation.
> > > >
> > > > **Q11:** Unanswered Questions in Supplementary Material
> > > >
> > > > > "In section D of in the supplementary material, two questions were not answered: Question 3 in D.5 and Question 4 in D.6."
> > > >
> > > > **A11:** We thank the reviewers for their careful review and apologize for the missed questions in the Supplementary Material - Data Sheet. To ensure completeness, we have re-examined and answered the two missed questions in Section D:
> > > >
> > > > D5Q3: What (other) tasks could the dataset be used for?
> > > >
> > > > D5A3: Visual anomaly detection and localization tasks under the PAD setting.
> > > >
> > > > D6Q4: Have any third parties imposed IP-based or other restrictions on the data associated with the instances?
> > > >
> > > > D6A4: No, the resources for all instances are open source.
> > > >
> > > > **Q12:** Implementation and Training Details
> > > >
> > > > > "The authors mention implementation and training details, though not for all evaluated methods. They write that code will be publicly available in a GitHub repository. At the time of writing this review, the linked  repository did not yet contain the all necessary information. If the missing parts are added upon publication, reproducing the results should be possible."
> > > >
> > > > **A12:** We value reviewer feedback on implementation and training details and are promoting open-sourcing of the code. Our MAD dataset is structurally almost identical to the widely used MvTec-AD dataset, with additional pose information being optional in the .json file.
> > > >
> > > > Thus, the implementation of the 10 baseline methods requires only modifications to the dataloader, and the code used for the benchmarks is attached to Github.
> > > >
> > > > Regarding the implementation of OmniposeAD, in addition to the description in Section 4.2, we have ensured that all necessary code will be publicly accessible in our GitHub repository in a few days.
> > > >
> > > > In order to make it possible to reproduce our methodology just by reading the paper, we have slightly modified the methodology section by adding more details and a clear explanation of the three modules in revised version.
> > > >
> > > > **Q13:** Discussion of Performance Degradation
> > > >
> > > > > "In Section 5.5 the authors write that the results in Table 3 demonstrate 'only marginal degradation in performance as the amount of training  data decreases'. In my opinion, some of the differences are more than  marginal decreases, for example 0.931 to 0.803 image-level AUROC for the object 'Puppy'."
> > > >
> > > > **A13:** We revisited the analysis in Section 5.5 to provide a more accurate description of the observed changes, ensuring that performance degradation is accurately reported and further explained:
> > > >
> > > > "As shown in Table.3: Despite the sparse training data leads to a slight decrease in performance, the less pronounced drop in pixel-level performance suggests that OmniposeAD excels in capturing local consistency within objects. The more significant decrease at the image-level indicates that sparse data impacts OmniposeAD's ability to learn the holistic normal features of object from global perspective."

---

> > > > > ### Author Response · Authors · 2023-08-21
> > > > > **Response to Reviewer upSJ (5 of 5).**
> > > > >
> > > > > **Q14:** Correctness of Table 2
> > > > >
> > > > > > "In Table 2, the bolded numbers are not always the highest ones in the  respective category. For example, the pixel score for "bird" is higher for OCRGAN than for "Ours", but the latter is bolded. These issues should be fixed for the final submission. (This does not change the general conclusions that can be drawn from the table.)"
> > > > >
> > > > > **A14:** We thank the reviewers for meticulously pointing out the inconsistencies in Table 2. To ensure accuracy and uniformity, we have corrected the discrepancies in bold in the revised version.
> > > > >
> > > > > **Q15:** Addressing Typos and Errors
> > > > >
> > > > > > "I found a few typos:  ● in the legend of Table 1: "comples" -> "complex"? ● in the legend of Figure 5: "performence" -> "performance"? ●in line 282 and 289: "pose diagnostic" -> "pose agnostic"?  ● Should the object category "pheonix" be named "phoenix"? The authors might want to check if there are any further typos before submitting the final version of the paper."
> > > > >
> > > > > **A15:** We place great value on the reviewers' meticulous identification of typos and errors. We thoroughly reviewed the manuscript and corrected any typos that were raised to ensure that the final version submitted was concise and error-free.
> > > > >
> > > > > We thank the reviewer again for recognizing our work and suggesting improvements; the revised version has been updated for the reviewers' reference and we look forward to further discussions.
> > > > >
> > > > > Authors

---

> > > > > > ### Comment · Reviewer_upSJ · 2023-08-22
> > > > > > **Response to the authors**
> > > > > >
> > > > > > Thank you very much for the very detailed response. It addressed a lot of my concerns.
> > > > > >
> > > > > > I have a follow-up comment to some of your replies.
> > > > > >
> > > > > > **Reply to A1 and A2:**
> > > > > >
> > > > > > While I still think that a more diverse background or objects with more variation would greatly enhance the dataset, I think the current setting is acceptable for a dataset that introduces a new problem setting (multi-pose AD). Thank you also for providing the "in-the-wild" images for illustration.
> > > > > >
> > > > > > **Reply to A6:**
> > > > > >
> > > > > > Thank you for the explanation. A clarifying question: If I understand it correctly, the steps you use to compute the color contrast are not explicitly described in the references you provided. Instead, the decision to use these steps is guided by the information in the references. Is my understanding correct? (To be clear: This is not a major issue of the paper.)
> > > > > >
> > > > > > **Reply to A7:**
> > > > > >
> > > > > > Thank you for the explanation. I may have misunderstood the definition of Segmentation Pixel-AUROC. Is the bounding box used for this metric the bounding box of the object or the bounding box of the segmented anomalous region?
> > > > > >
> > > > > > **Reply to A10:**
> > > > > >
> > > > > > Thank you for the further information. I look forward to seeing the MAD-Real dataset when it is released. In case the review of MAD-Real is finished in time, it would be great if more information like example images could be added to the supplementary material.
> > > > > >
> > > > > > **Reply to A14:**
> > > > > >
> > > > > > Just for completeness: You missed a few numbers. The scores for OCRGAN are higher than the ones for "Ours" for the objects "Sheep", "Scorpion" and "Obesobeso".

---

> > > > > > > ### Author Response · Authors · 2023-08-23
> > > > > > > **Thank you for your response!**
> > > > > > >
> > > > > > > Dear Reviewer upSJ,
> > > > > > >
> > > > > > > Thank you again for recognizing our work and discussing it further, and here we would like to respond your concerns and share our plans for official release.
> > > > > > >
> > > > > > > **Q1:**
> > > > > > >
> > > > > > > > "While I still think that a more diverse background or objects with more variation would greatly enhance the dataset, I think the current setting is acceptable for a dataset that introduces a new problem setting (multi-pose AD). Thank you also for providing the "in-the-wild" images for illustration. "
> > > > > > >
> > > > > > > **A1:** Thank you for your recognition for the MAD dataset and further suggestions. All the authors agree on your point about the diversity of the dataset.
> > > > > > >
> > > > > > > We'd be happy to share with you more recent attempts to add diversity to the backgrounds of objects. For MAD-Sim, each sample is actually rendered from Blender as we described. We tried to add complex backgrounds (real photographs and some of Blender's built-in textures), but the visual results were quite poor, e.g. the objects stayed in high resolution but the backgrounds were blurry. The most important point is that during the MAD-Sim data generation process, we rotate the object itself to collect multi-pose samples, and if we add a complex background (which is more like a fixed 2D backscene), it is hard to keep the samples reasonable, i.e., if the object changes its pose, the background will also change accordingly. But this problem is solved naturally in MAD-Real, we just need to pick up the camera.
> > > > > > >
> > > > > > > We have a compromise solution: our final version will no longer change the content of the MAD-Sim dataset (as we know many researchers have already tried). However, we can open-source the Blender model files for the 20 Legos used in MAD-Sim to provide a pipeline for changing the backgrounds, making it easy for researchers who wish to explore more avenues to try.
> > > > > > >
> > > > > > > Do the reviewer think this is OK?
> > > > > > >
> > > > > > > **Q2:**
> > > > > > >
> > > > > > > > "Thank you for the explanation. A clarifying question: If I understand it correctly, the steps you use to compute the color contrast are not explicitly described in the references you provided. Instead, the decision to use these steps is guided by the information in the references. Is my understanding correct? (To be clear: This is not a major issue of the paper.)"
> > > > > > >
> > > > > > > **A2:** Thank you for further discussion. Your understanding is absolutely correct. We hope to explore the connection between object attributes and AD algorithm performance in a way that is as consistent with simple human intuition of judgment as possible.
> > > > > > >
> > > > > > > **Q3:**
> > > > > > >
> > > > > > > > "Thank you for the explanation. I may have misunderstood the definition of Segmentation Pixel-AUROC. Is the bounding box used for this metric the bounding box of the object or the bounding box of the segmented anomalous region?"
> > > > > > >
> > > > > > > **A3:** Thank you for your follow up question. The bounding box used for this metric is the object's bounding box. We want to ignore any misjudgment of the background outside of the object. Those misjudgments can drastically affect the image-level anomaly classification results.
> > > > > > >
> > > > > > > **Q4:**
> > > > > > >
> > > > > > > > "Thank you for the further information. I look forward to seeing the MAD-Real dataset when it is released. In case the review of MAD-Real is finished in time, it would be great if more information like example images could be added to the supplementary material."
> > > > > > >
> > > > > > > **A4:** Thank you for your kind attention to our dataset! We will open source the MAD-Real dataset at the first opportunity. We'll make sure to add more Real dataset information to the supplementary material before camera-ready.
> > > > > > >
> > > > > > > **Q5:**
> > > > > > >
> > > > > > > > "Just for completeness: You missed a few numbers. The scores for OCRGAN are higher than the ones for 'Ours' for the objects 'Sheep', 'Scorpion' and 'Obesobeso'. "
> > > > > > >
> > > > > > > **A5:** Thank you for your careful review, we have revised the bolding of the table data. We will release the latest revision in the near future.
> > > > > > >
> > > > > > >
> > > > > > > Authers

---

> > > > > > > > ### Comment · Reviewer_upSJ · 2023-08-23
> > > > > > > > **Reply to the authors**
> > > > > > > >
> > > > > > > > **Reply to A1:**
> > > > > > > > > We have a compromise solution: our final version will no longer change the content of the MAD-Sim dataset (as we know many researchers have already tried). However, we can open-source the Blender model files for the 20 Legos used in MAD-Sim to provide a pipeline for changing the backgrounds, making it easy for researchers who wish to explore more avenues to try.
> > > > > > > > >
> > > > > > > > > Do the reviewer think this is OK?
> > > > > > > >
> > > > > > > > I think that is OK. Thank you for the explanation concerning the difficulty of using the background in the multi-pose setting.
> > > > > > > >
> > > > > > > > **Reply to A3:**
> > > > > > > >
> > > > > > > > Thank you for the clarification. Using the bounding box of the object is much more reasonable than my initial understanding that the bounding box of the segmented region is used. This addresses my initial concern. I think that false positive anomaly detections in the background should not be completely ignored in an evaluation. However, since the Segmentation Pixel-AUROC is an additional metric that can complement other metrics which incorporate the background, that is okay in this case.

---

### Official Review · Reviewer_9dF6 · 2023-07-20
**can be a good work after major revision**

**Rating:** 6
**Confidence:** 5
**Correctness:** appropriate
**Clarity:** yes

**Strengths:**

1. the quality of the newly introduced dataset is high in terms of outlier type, data size, etc. More important, it is a unique dataset about pose-specific anomalies.

2. comprehensive experiments with 10 baselines ( most of which were within recent three years) can serve as a benchmark.

3. the proposed PAAD outperformed all baselines



**Additional Feedback:**

no

**Documentation:**

yes

**Limitations:**

yes but not enough

**Opportunities For Improvement:**

1. there were no any baseline methods published before 2019 (see below statistics), which was not reasonable.

[14,26] 2019
[15,19,30,50] 2022
[19] 2022
[20,39,44,52]2021

2. the limitation of the proposed framework needs more discussion.

3. since the experiments were only based on the newly introduced dataset in this paper, the potential risk of the conclusion obtained should be addressed.

4. lack of necessary typical references most of the references (>90%) were after 2020.




**Relation To Prior Work:**

yes

**Summary And Contributions:**

this study was about pose-agnostic anomaly detection with three main contributions:
(1) introduced a new dataset
(2) conducted comprehensive experiments
(3) introduced a framework for pose-agnostic anomaly detection

---

> ### Author Response · Authors · 2023-08-18
> **Response to Reviewer 9dF6 (1 of 2).**
>
> We thank reviewer's insightful evaluation of manuscript and the valuable feedback provided. Your detailed comments have been immensely beneficial, and we are committed to addressing your concerns in order to enhance the quality of our work. We have taken your feedback into serious consideration and have outlined our responses below:
>
> **Q1**: Baseline Methods Selection and Explanation
>
> > "there were no any baseline methods published before 2019 (see below statistics), which was not reasonable."
>
> **A1:** We thank the reviewers for the review and attention to the selection of the baseline methods, appropriate baseline methods are vital for evaluating PAD setting. To ensure that the selection process was reasonable, in addition to the time when the methods were proposed, we considered the following aspects:
>
> 1) **Representative.** We evaluate the PAD setup on the MAD dataset from eight different anomaly detection paradigms: Teacher-Student Architecture/One-Class Classification (OCC)/Distribution-Map/Memory Bank/ Autoencoder (AE)/Generative Adversarial Networks (GANs) /Transformer/ Pseudo-anomaly. For each paradigm we chose 1 or 2 relatively representative ones as baselines.
>
> 2) **Performance Advancement.** For each paradigm, we try to select methods with sota performance as much as possible, which are usually proposed relatively recently.
>
> 3) **Reproducibility.** In order to avoid detailed ambiguities in the understanding of the original method, we only consider methods for which there is released official code.
>
> 4) **Research trends.** The anomaly detection community typically recognizes the MvtecAD dataset (proposed in 2019) as an important node that has largely enabled the development of deep learning-based industrial anomaly detection algorithms. Moreover, this dataset is the first anomaly detection dataset centered on objects (and, of course, a portion of textures) that contain a wide range of color RGB.
>
> The above is what produces the coincidence of the selected baseline methods after 2019. We would be open to further guidance from reviewer on the reasonability of baseline selection.
>
>
>
> **Q2:** Further Discussion of Ours Framework Limitations
>
> > "the limitation of the proposed framework needs more discussion. "
>
> **A2:** We appreciate the reviewer's insight regarding the limitations of our proposed framework (In the revised version we changed the "framework" to more understandable term "OmniposeAD"). In the original paper, we briefly presented several limitations of the framework due to space constraints, but we understood that more detailed discussion of the limitations was essential to our work. In response, we revised the manuscript and expanded the discussion of the limitations of OmniposeAD, pointing out potential avenues for future improvement:
>
> "For our proposed OmniposeAD, it is difficult for generalization to novel categories to obtain appreciable performance; for all modules we chose the vanilla method aimed at avoiding additional ablation experiments, so with the rapid development of NeRF tech, it has the potential to achieve faster training times, less training data, and more realistic high-frequency details."
>
> Details can be found in the revised version of Sec.6.

---

> > ### Author Response · Authors · 2023-08-18
> > **Response to Reviewer 9dF6 (2 of 2).**
> >
> > **Q3:** Addressing Potential Conclusion Risks
> >
> > > "since the experiments were only based on the newly introduced dataset in this paper, the potential risk of the conclusion obtained should be addressed."
> >
> > **A3:** We thank reviewer for his concerns and feedback on the conclusions of our paper. The reviewer mentioned that our experiments were based on a single dataset only, and we truly understand the concerns about the potential risks associated with the conclusions. We hope to address the reviewers' concerns with the following discussion:
> >
> > **Why did we only experiment on the MAD dataset?**
> >
> > We introduce a challenging and novel anomaly detection setting: PAD. To the best of our knowledge existing datasets (lacking multi-pose RGB information and pose information) are not suitable for fair evaluation of the PAD setting, so we develop the MAD dataset to fill this gap of the study. This is the reason why we performed our experiments only on the newly proposed dataset.
> >
> > **How do we avoid potential risks to our conclusions?**
> >
> > 1) We have novel but well-designed MAD datasets to fully evaluate the PAD setup. Our dataset consists of 20 classes of LEGO toys of different shapes and colors, and three different defects, with 4K+ sampled views and 11,000+ samples with pixel-accurate GT. Beyond that, we developed simulated/real datasets that make the evaluation of the method highly robust.
> >
> > 2) We carefully selected representative benchmark methods and ensured consistent data preprocessing pipeline and hyperparameter selection during the experiments.
> >
> > 3) The experimental results are consistent with human intuition. For example, the results are relatively better on the categories *cat* and *obesobeso*, which means that the algorithm is more likely to capture structural information on simple shaped objects; the relatively poor results on the categories *turtle* and *unicorn* imply that the algorithm has difficulty in capturing discriminable texture information for objects with a single color.
> >
> > Again, we thank the reviewers for their interest in the conclusions, and we look forward to further discussion in avoiding potential risks to the conclusions!
> >
> >
> >
> > **Q4:** Inclusion of Necessary References
> >
> > > "lack of necessary typical references most of the references (>90%) were after 2020."
> >
> > **A4:** We thank the reviewers for suggesting the lack of typical literature citations. After careful checking and counting, we found that there were 56 citations in the initial version of the submission, of which 47 citations, or **84%, were indeed after 2020, rather than >90% as mentioned by the reviewer.** In fact, there are 9/56 papers cited before 2020: [4]2019, [14]2019, [18]2015, [23]2015, [26]2019, [28]2018, [35]2015, [38]2019, [42]2017.
> >
> > We agree with the importance of citing typical literature and add to the following:
> >
> > **In the dataset scope:** We have fully investigated the typical object anomaly detection datasets and refer to the latest comprehensive survey"Deep Industrial Image Anomaly Detection: A Survey (https://arxiv.org/abs/2301.11514?file=2301.11514)". We believe the earliest object anomaly detection dataset is GDXray (2015), which we have cited.
> >
> > **In the AD method scope:** In fact we considered references to typical (as early as possible) methods e.g., for the classical method: [5] (2020), [50] (2020), [38] (2021), [31] (2022), [47] (2021), [46] (2021), [26] (2021). According to our knowledge, there are indeed many classical methods (<2018) for anomaly detection, but they focus on surface anomaly detection and cannot take into account object anomaly detection.
> >
> > Nevertheless, we have selected the most relevant and typical literature for citation in revised version: In GDXray dataset, we added a classcial method[15] (2018); GANs: schlegl2017unsupervised and schlegl2019f; AE: baur2019deep; Distribution map: sabokrou1609fully.
> >
> > We are open to add any typical literature on object anomaly detection recommended by the reviewers.
> >
> > We thank the reviewer again for approving our work and suggesting improvements, revised version will be updated in the near future and we look forward to any further discussions.
> >
> > Authors

---

> > ### Comment · Reviewer_9dF6 · 2023-08-22
> > **Thanks for your reply**
> >
> > "we only consider methods for which there is released official code"
> > However, this is not a valid reason for the guidance of baseline selection, preventing this work to be better.

---

> > > ### Author Response · Authors · 2023-08-23
> > > **Thank you for your response!**
> > >
> > > Dear reviewer 9dF6,
> > >
> > > Thanks again for the further discussion and for pointing out the remaining concerns. We will include the limitations you pointed out regarding the choice of baseline methods for benchmark as one of the limitations in the final version. We look forward to more constructive and detailed discussions on the MAD dataset and the OmniposeAD method.
> > >
> > > Authors

---

### Official Review · Reviewer_ryha · 2023-07-21
**The paper is all right, but there is room for improvement.**

**Rating:** 8
**Confidence:** 5
**Correctness:** Yes, I find the claims made in the pa…
**Clarity:** Yes, the paper is well-structured and…

**Strengths:**

1. The authors present a novel perspective on the object anomaly detection task through PAD, along with a well-designed and user-friendly dataset, MAD. This contribution brings new challenges to the field and bridges the gap between research and practical applications, surpassing existing 2D/3D anomaly detection settings.
2. The benchmarking approach, considering different AD paradigms, provides valuable insights for the application community to select appropriate AD methods. The inclusion of object attribute quantification as part of the benchmark enhances the practicality and relevance of the evaluation.
3. The utilization of NeRF as an anomaly detection method in PAD is a novel contribution. The authors demonstrate its effectiveness through promising performance results.

**Additional Feedback:**

See Above.

**Documentation:**

The authors have adequately addressed hosting plans, long-term data accessibility, and provided a datasheet for the datasets. The MAD-Sim dataset's URL is accessible, and the authors commit to open-sourcing the MAD-Real challenge dataset and code. The provided documentation meets the requirements of the D&B track.

**Limitations:**

While PAD demonstrates generalization capabilities for unseen views/angles of objects, it may face challenges when generalizing to new object categories. However, PAD remains a robust and applicable solution for industrial quality control and related applications.

**Opportunities For Improvement:**

To improve readability and provide more insightful analysis, I suggest the following revisions:
1. Provide a concise description of the author's contributions in Section 1 (lines 58-77).
2. Consider using more refined terminology for object attributes, such as "structure" instead of "shape" and "texture" instead of "color." Additionally, provide a deeper explanation of the practical implications of quantifying attributes, going beyond the mere description of the quantification steps.
3. While the paper benchmarks PAAD with various AD paradigms, I suggest conducting further analysis on the quantitative results of AD and anomaly localization. This would help uncover the factors contributing to the varying performance of different methods under different AD paradigms within the PAAD setting.
4. Is the concept of PAD the same with PAAD? I suggest the author use the uniform one.
5. Incorporate the checklist into the main paper rather than placing it in the supplementary material.

**Relation To Prior Work:**

Yes, the authors provide a detailed discussion of related works, as evident in Table 1 and the related works section.

**Summary And Contributions:**

This paper proposed PAD to tackle two problems: 1) Exisiting AD datasets lacks comprehensive visual information from multiple pose angles. They usually have an unrealistic assumption that the anomaly-free training dataset is pose-aligned, and the testing samples have the same pose as the training data. 2) There have absence of a consensus on experimental settings for pose-agnostic anomaly detection leads to unfair comparisons of different methods, hindering the research on it. The authors 1) built a novel dataset: MAD, which containing 20 Lego animals, as well as images and pose information sampled from multiple perspectives, for the PAD problem. 2) provide across-the-board benchmark tests in anomaly detection, localization and object attributes, respectively. 3) proposed PAAD framework to learn the full representation of normal objects using NeRF, achieving satisfactory performance under the PAD setting.

---

> ### Author Response · Authors · 2023-08-18
> **Response to Reviewer ryha (1 of 2).**
>
> We would like to thank the reviewer for the thorough review of our manuscript and for providing the constructive feedback to improve the quality. We highly value the reviewers' suggestions and have addressed each point raised as follows:
>
> **Q1:** Concise the Description of Author's Contributions
>
> > "Provide a concise description of the author's contributions in Section 1 (lines 58-77)."
>
> **A1:** We appreciate reviewer's observation regarding the description of the author's contributions in Section 1. In response, we have revised and condensed our contribution in revised version to ensure a concise and comprehensive overview of our 4 research objectives and contributions:
>
> - We introduced Pose-agnostic Anomaly Detection (PAD), a challenging setting for object anomaly detection from diverse viewpoints/poses, breaking free from the constraints of stereotypical pose alignment, and taking step forward to pratical anomaly detection and localization tasks.
> - We developed the Multi-pose Anomaly Detection (MAD) dataset, the first attempt at evaluating the pose-agnostic anomaly detection. Our dataset comprises MAD-Sim and MAD-Real, containing 11,000+ high-resolution RGB images of multi-pose views from 20 diverse shapes and colors of LEGO toys, offering pixel-precise ground truth for 3 types of anomalies.
> - We conducted a comprehensive benchmark of the PAD setting, utilizing the MAD dataset. Across 8 distinct paradigms, we meticulously selected 10 state-of-the-art methods for evaluation. In a pioneering effort, we delving into the previously unexplored realm of correlation between object attributes and anomaly detection performance.
> - We proposed OmniposeAD, utilizing NeRF to encode anomaly-free object attributes from diverse viewpoints/poses and comparing reconstructed normal reference with query image for pose-agnostic anomaly detection and localization. It outperforms previous methods quantitatively and qualitatively on the MAD benchmark, which indicates the promising potential on PAD setting.
>
> **Q2:** Refine the Terminology for Object Attributes and Practical Implications
>
> > "Consider using more refined terminology for object attributes, such as 'structure' instead of 'shape' and 'texture' instead of 'color.' Additionally, provide a deeper explanation of the practical implications of quantifying attributes, going beyond the mere description of the quantification steps."
>
> **A2:** We have thoroughly considered reviewer's suggestion regarding the use of term for object attributes in our work. After a discussion among the authors, it was decided to replace "shape" with "structure", but to keep "color" instead of "texture". Here we discuss the reasons behind the choice of terminology and provide a rationale for our current decision:
>
> Shape vs. Structure: The term "shape" was initially considered because it is an easily understood concept, and people can't help but perceive the simple external shape of an object when they are handed it. After our discussion, we further realized that the geometry and arrangement of the parts of an object are important (Lego toys, as well as most industrially produced objects, are mostly assembled from parts). In the field of computer vision and object recognition, "structure" is usually a high-level concept that takes into account intrinsic connections further than considering the "shape" of an object. We therefore believe that the use of the term "structure" is more appropriate; Color vs. Texture: Distinguishing between "color" and "texture" is consistent with our perception of the visual attributes associated with objects. "Color" refers primarily to the color information (richness/complexity) of an object, while "texture" includes specific patterns and regular visual information about the surface of an object. "Texture" represents the visual effect of the surface in addition to the tactile attributes, which are usually not perceived at RGB resolution. By using more precise "color" terminology, we can ensure that readers can easily understand the specific attributes we are referring to without creating further ambiguity. Therefore, we prefer to keep the term "color".
>
> We would be happy to have further discussions with reviewer regarding attribute terminology.
>
> **Why introduced quantitative object attributes to evaluate performance?**
>
> Anomaly detection algorithms are evolving rapidly, but it is often difficult to intuitively determine which one is the best from performance metrics when trying to deploy an algorithm to a real scenario. To reduce testing costs, we introduce the most intuitive object attributes: structure and color. When we are given an object to be tested, it is easy to determine which algorithm may be optimal by referring to the correlation between the quantitative metrics of the object attributes and the performance of different algorithms.

---

> > ### Author Response · Authors · 2023-08-18
> > **Response to Reviewer ryha (2 of 2).**
> >
> > **Q3:** More Analysis on Quantitative Results.
> >
> > > "While the paper benchmarks PAAD with various AD paradigms, I suggest conducting further analysis on the quantitative results of AD and anomaly localization. This would help uncover the factors contributing to the varying performance of different methods under different AD paradigms within the PAD setting."
> >
> > **A3:** We thank the reviewer for the attention and comments on the quantitative results analysis section. We have provided streamlined but insightful analysis results in the experimental section due to page limitations. We are certainly happy to share more details of our findings:
> >
> > **OmniposeAD's Performance and Data Enhancement:**
> >
> > Experiments compare OmniposeAD with leading methods for pose-agnostic anomaly detection (PAD), showing OmniposeAD's superior performance on MAD dataset. It outperforms CFlow and UniAD significantly (+7.0/+19.6 and +8.7/+28.7, respectively), especially due to using supplementary data like pre-trained ImageNet. In contrast, methods like DRAEM and Cutpaste using pseudo-anomalies for augmentation struggle in the PAD context due to diverse object representations across poses.
> >
> > **Potential of Embedding and Reconstruction in PAD:**
> >
> > Feature embedding and reconstruction methods, like STFPM, CFlow, CFA, FAVAE, and UniAD, perform well in pixel-level anomaly localization (with slight fluctuations). However, besides OmniposeAD, benchmark methods fare poorly in image-level anomaly detection. Even UniAD struggles with multi-pose anomalies. Addressing this requires exploring algorithms adept at capturing normal attributes of multi-pose objects for robust image-level anomaly detection in PAD.
> >
> > **Effect of Dense-to-Sparse training data on OmniposeAD**:
> >
> > We assess the impact of transitioning OmniposeAD from dense to sparse viewpoint training data using the MAD-Real dataset. We train OmniposeAD with limited samples (100, 70, and 50) to reflect challenges in obtaining dense viewpoint data. Results in Table.3 show that sparse training data leads to slight performance reduction. Pixel-level performance decline suggests OmniposeAD's strength in local consistency capture, while image-level drop indicates sparse data's impact on holistic object feature learning from a global perspective.
> >
> > More details reviewer can refer to Sec.5.3 and 5.5 in revised version.
> >
> >
> >
> > **Q4:** Clarification of PAD and PAAD Terminology
> >
> > > "Is the concept of PAD the same with PAAD? I suggest the author use the uniform one."
> >
> > **A4:** Reviewer's query regarding the concept "PAD" and "PAAD" has been noted. In fact, they refer to distinct concepts. To express our work more clearly, we have decided not to continue to use the term 'PAAD', and further introduce more appropriate definitions:
> >
> > PAD - introduced new anomaly detection setting; MAD - developed new anomaly detection dataset for PAD evaluation; OmniposeAD - proposed new NeRF-based method for PAD.
> >
> > Note that the proposed new definition has been replaced in the revised version.
> >
> > **Q5:** Checklist-from Supple. to Main Paper
> >
> > > "Incorporate the checklist into the main paper rather than placing it in the supplementary material."
> >
> > **A5:** Thank you for your comments. We have integrated the checklist into the main body of the revised version.
> >
> > We thank the reviewer again for approving our work and suggesting improvements, revised version will be updated in the near future and we look forward to any further discussions.
> >
> > Authors,

---

> > > ### Comment · Reviewer_ryha · 2023-08-21
> > > **Discussion feedback**
> > >
> > > Thanks to the authors for the response and clarifications. I am slightly more positive about the paper after the author have improved the paper (including improvements suggested by others) and clarified some misunderstandings on my part about the concepts.
> > >
> > > Some additional discussion of the responses:
> > >
> > > - For A2, I was only suggesting but not intending to ask for a change in the terms used for object attributes. But the author's explanation sounds plausible.
> > > - For A4, I agree that the authors provide clearer definitions to separate the ambiguous meanings of PAAD and PAD.
> > >
> > > I think that, overall, the addition of the quantitative results analysis compensates for the superficial discussion of PAD and provides a clearer description of the main contributions, making this paper convincing enough to demonstrate the significance of proposing a PAD setting and a MAD dataset. I therefore raise my rating to 8: clear accept.

---

> > > > ### Author Response · Authors · 2023-08-23
> > > > **Thank you for your response!**
> > > >
> > > > Dear reviewer ryha,
> > > >
> > > > Thank you again for recognizing our work and further discussion. We will select the most appropriate terms for object attributes in the final version.
> > > >
> > > > Authors

---

### Official Review · Reviewer_pCfQ · 2023-07-21
**Significant PAD Setting with Dataset and Benchmarking for Object Anomaly Detection Task**

**Rating:** 8
**Confidence:** 4

**Strengths:**

- The proposed pose-agnostic anomaly detection setting addresses a real and intriguing problem. Anomaly detection on the production line does not guarantee that objects will consistently appear with fixed poses in the camera view. Consequently, objects with defects in unobserved orientations are easily overlooked. This paper offers a promising approach to tackle this critical challenge.
- The authors present a well-designed and high-quality simulated+real dataset, incorporating RGB and pose information, to establish a benchmark for the PAD setting. It is worth noting that the file structure of the MAD dataset aligns with MvTec-AD, enabling easy replication of the benchmarking process and fostering the development of new algorithms.
- The PAD benchmark evaluates multiple algorithmic paradigms, facilitating reproducibility and providing fresh insights through the analysis of quantified object attributes.
- The authors propose a novel NeRF-based object anomaly detection framework for the PAD setting, exhibiting exceptional performance.

**Additional Feedback:**

Please refer to the "Opportunities For Improvement" and "Limitations" sections for additional feedback.

**Clarity:**

Yes, the paper is well-presented and easy to follow. The logical flow of the paper is coherent.

**Correctness:**

Yes, the claims made in the submission are accurate. The dataset construction is methodical, and the benchmark methods, metrics, and experimental design are appropriate.


**Documentation:**

Yes, the paper provides sufficient detail on data collection and organization, availability and maintenance, and ethical and responsible use. The MAD-Sim dataset is available on GitHub under the CC BY 4.0 license. The implementation of benchmarking methods is also provided on GitHub. The authors explicitly state that the code for PAAD and the MAD-Real set will be open source. To the best of my knowledge, the MAD-Sim dataset is indeed adequate for exploring the PAD setting, which is commendable.


**Ethics:**

I have found no significant ethical concerns regarding the submitted content.

**Limitations:**

The authors have acknowledged the limitations in the conclusion section.

**Opportunities For Improvement:**

The paper could be further enhanced in the following aspects:
- While Fig. 1(b) graphically represents the PAD setting, it would be beneficial to provide a rigorous mathematical notation to precisely define the setting.
- In Table 1, the abbreviated letter comments could be placed in the footnotes. Additionally, please review the color attribute descriptions "Syn." for the Eyecandies dataset.
- Although adhering to strict page limits, the paper in the D&B Track should include more dataset-related figures. Noteworthy figures showcased in the supplemental material and the Github Repository, such as defect samples and performance visualizations, could be selectively incorporated into the main paper. Furthermore, there is potential for abbreviation in author information and the structure of the PAAD figure.

**Relation To Prior Work:**

Yes, the differences between MAD and previously related datasets are clearly demonstrated in Table 1.

**Summary And Contributions:**

This paper presents a novel pose-agnostic anomaly detection (PAD) setting and introduces a comprehensive object anomaly detection dataset (MAD) comprising multiple views of objects. The dataset includes quantitative object attributes for 20 classes of Lego toys, facilitating in-depth analysis of algorithm performance. Additionally, the authors propose a NeRF-based anomaly detection framework (PAAD) as a baseline for the PAD setting. The value of the dataset is demonstrated by benchmarking a diverse range of methods under various algorithm paradigms for anomaly detection and localization.

---

> ### Author Response · Authors · 2023-08-18
> **Response to Reviewer pCfQ.**
>
> We thank reviewer's thorough review of our manuscript and the insightful feedback provided. Reviewer's suggestions are highly valued, and we have taken them into account, addressing each point raised as outlined below:
>
> **Q1:** Rigorous Mathematical Notation for the PAD Setting.
> > "While Fig.1(b) graphically represents the PAD setting, it would be beneficial to provide a rigorous mathematical notation to precisely define the setting."
>
> **A1:** As mentioned by the reviewer, we use illustration to explain the PAD setting in our paper. To ensure there is no ambiguity, we further introduce precise notation definitions to describe our proposed PAD setting in Sec. 4:
>
> "Our Pose-agnostic Anomaly Detection (PAD) setting introduced for object anomaly detection and localization tasks shown as Figure.1 and it can be formally stated as follows: Given a set of training examples $ \\mathcal{T}= \\left \\{t_{i} \\right \\}_{i=1}^{N}$, in which $\\left\\{t_1, t_2, \\cdots, t_N\\right\\}$ are the anomaly-free samples from object's multi pose view and each $t$ consists of a RGB image $I _{r g b}$ with pose information $\\theta _{pose}$. In addition, $\\mathcal{T} _{n}$ belongs to a certain object $o _{j}$, $o _{j}\\in\\mathcal{O}$, where $\\mathcal{O}$ denotes the set of all objects categories. During testing, given a query (normal or abnormal) sample $\\mathcal{Q}$ from object $o _{j}$ without pose information $\\theta _{pose}$, the pre-trained AD model $M$ should discriminate whether or not the query sample is anomalous and localize the pixel-wise anomaly region if the anomaly is detected.
>
> **Q2:** Footnotes for Table.1 Captions and Color Attribute Descriptions.
>
> > "In Table 1, the abbreviated letter comments could be placed in the footnotes. Additionally, please review the color attribute descriptions 'Syn.' for the Eyecandies dataset."
>
> **A2:** We thank the reviewer's careful review of Table.1 and suggestions for improvement.
> 1) To enhance the readability of Table.1, we have placed the details of the interpretation of the abbreviated letters in footnote 1.
> 2) Furthermore, we checked the Eyecandies dataset for the correctness of the color attribute descriptions, and 'Syn.' should indeed be corrected to 'Diversity'.
>
> Reviewer can refer to Table.1 in the revised version.
>
> **Q3:** Add More Dataset-Related Figures.
>
> > "Although adhering to strict page limits, the paper in the D&B Track should include more dataset-related figures. Noteworthy figures showcased in the supplemental material and the Github Repository, such as defect samples and performance visualizations, could be selectively incorporated into the main paper. Furthermore, there is potential for abbreviation in author information and the structure of the PAAD figure."
>
> **A3:** We deeply value your suggestion to amplify the inclusion of dataset-related visuals within the main manuscript. Actually we did prepare very rich graphical representations of the dataset, but were limited by page space and had to put in supplementary material. Based on the reviewers' comments for improvement, we selectively added more necessary illustrations in the revised version:
>
> 1) Illustrations of the 20 categories of LEGO toys in the MAD dataset (Figure.2(a))
>
> 2) Example of the three defect types included in the MAD dataset (Figure.2(b))
>
> 3) Updated and added GT illustrations for the visualization results (Figure.4)
>
> We hope that the added illustrations will help anyone understand the MAD dataset better, and we welcome further comments from reviewers.
>
> **Additional Improvement:** Based on the reviewers' comments, we realized that there was room to reduce space in the main manuscript, such as author information and few illustrations, and we have made appropriate changes.
>
> We thank the reviewer again for approving our work and suggesting improvements, revised version will be updated in the near future and we look forward to any further discussions.
>
> Authors，

---

> > ### Comment · Reviewer_pCfQ · 2023-08-22
> >
> > Thank you to the authors for their responses. After reviewing the latest updated version of the manuscript, I think there is one more point that the authors need to be aware of: some notations in Sec.4.2 should be aligned with those defined in Sec4.1. Then I believe that most of the concerns I raised were properly addressed. PAD setting would provide anomaly detection community with the opportunity to be more flexible and closer to real-world applications, so I tend to stick with my original rating.

---

> > > ### Author Response · Authors · 2023-08-23
> > > **Thank you for your response!**
> > >
> > > Dear reviewer pCfQ,
> > >
> > > Thank you again for recognizing our work and detailed comments.We will ensure notational consistency between Sec4.1 and Sec4.2 in the final version.
> > >
> > > Authors

---

### Author Response · Authors · 2023-08-21
**Looking forward to further discussion.**

Dear Reviewers,

Thank you again for your constructive reviews, which have helped us improve the quality and clarity of the paper! Up to now, we have updated our responses to each of your comments in your thread and uploaded the latest revised version of the changes made in response to the comments for the reviewers' information and further discussion. We hope that we have been able to address your concerns. As we approach the end of the discussion period approach, please don’t hesitate to let us know if you have any additional questions or comments. We look forward to the discussion!

Thanks for your time,

Authors

---

### Decision · Program_Chairs · 2023-09-22

**Decision:**

Accept (Poster)

**Comment:**

The paper received four reviews. The authors submitted rebuttals and the reviewers engaged in conversation with the authors, who clarified the raised weaknesses. The reviewers appreciate the well-designed and high-quality simulated+real dataset, which addresses a real and intriguing problem. The reviewers also consider that the utilization of NeRF as an anomaly detection method is a novel contribution. In summary, the reasons to accept the paper clearly outweigh the negative points. However, the authors are asked to include the relevant points mentioned in the discussion to the camera ready.